# A comprehensive study of genetic regulation and disease associations of plasma circulatory microRNAs using population-level data

Rima Mustafa[1,2,3], Michelle M. J. Mens[4,5], Arno van Hilten[6], Jian Huang[1,7,8], Gennady Roshchupkin[4,6], Tianxiao Huan[14,15], Linda Broer[9], Joyce B. J. van Meurs[9,10], Paul Elliott[1,2,11,12,13], Daniel Levy[14,15], M. Arfan Ikram[4], Marina Evangelou[16], Abbas Dehghan[1,2,11†] and Mohsen Ghanbari[4*†]

†Abbas Dehghan and Mohsen Ghanbari are joint senior authors.

*Correspondence:
m.ghanbari@erasmusmc.nl

⁴ Department of Epidemiology, Erasmus MC University Medical Center Rotterdam, Rotterdam, The Netherlands
Full list of author information is available at the end of the article

## Abstract

**Background:** MicroRNAs (miRNAs) are small non-coding RNAs that post-transcriptionally regulate gene expression. Perturbations in plasma miRNA levels are known to impact disease risk and have potential as disease biomarkers. Exploring the genetic regulation of miRNAs may yield new insights into their important role in governing gene expression and disease mechanisms.

**Results:** We present genome-wide association studies of 2083 plasma circulating miRNAs in 2178 participants of the Rotterdam Study to identify miRNA-expression quantitative trait loci (miR-eQTLs). We identify 3292 associations between 1289 SNPs and 63 miRNAs, of which 65% are replicated in two independent cohorts. We demonstrate that plasma miR-eQTLs co-localise with gene expression, protein, and metabolite-QTLs, which help in identifying miRNA-regulated pathways. We investigate consequences of alteration in circulating miRNA levels on a wide range of clinical conditions in phenome-wide association studies and Mendelian randomisation using the UK Biobank data (*N* = 423,419), revealing the pleiotropic and causal effects of several miRNAs on various clinical conditions. In the Mendelian randomisation analysis, we find a protective causal effect of miR-1908-5p on the risk of benign colon neoplasm and show that this effect is independent of its host gene (*FADS1*).

**Conclusions:** This study enriches our understanding of the genetic architecture of plasma miRNAs and explores the signatures of miRNAs across a wide range of clinical conditions. The integration of population-based genomics, other omics layers, and clinical data presents opportunities to unravel potential clinical significance of miRNAs and provides tools for novel miRNA-based therapeutic target discovery.

**Keywords:** MicroRNA, Expression quantitative trait loci, Population-based cohort

## Background

MicroRNAs (miRNAs) are small non-coding RNAs of approximately 22 nucleotides that regulate gene expression at the post-transcriptional level. They play critical roles in determining whether genes are (in)active and proteins are translated [1, 2]. Over 2500 high-confidence miRNAs have been identified in humans [3], which are predicted to regulate more than half of protein-coding genes through cleavage or translation repression of messenger(m)-RNAs [4, 5]. miRNAs have shown their potential as disease biomarkers [6] and, to a lesser extent, therapeutic targets [7]. Identification of the role of miRNAs in regulating the expression of specific genes and their effects in clinical conditions has been a subject of extensive work in recent years. However, the genetic regulation of miRNAs remains less well understood.

Circulating miRNAs are released from cells into circulation via extracellular vesicles such as exosomes [8]. The high reliability and stability as well as accessibility in blood make circulating miRNAs important candidates as diagnostic and prognostic biomarkers in human diseases. Genetic variants are known to regulate the level of miRNAs in the circulation [9–11] or tissues and cells [12–14], referred to as miRNA expression quantitative trait loci (miR-eQTLs). Previous studies on subset of miRNAs showed that miR-eQTLs contribute to a proportion of variation in miRNA levels [9, 10], with a tiny percentage of miR-eQTLs replicated across studies thus far [9]. The identified miR-eQTLs have been used also to study the effect of perturbation of miRNA levels on disease risk [9, 10, 14]. However, such an effect on a wide range of clinical conditions at the population level remains to be elucidated. Unravelling the genetic regulation of high-confidence miRNAs can provide insights into their roles in affecting disease risk and discover potential therapeutic targets.

This study measured plasma levels of 2083 circulating miRNAs in the population-based Rotterdam Study cohort using a targeted next-generation sequencing platform (HTG EdgeSeq miRNA Whole Transcriptome Assay), which allows simultaneous, quantitative detection of miRNAs with a high sensitivity and specificity [15, 16]. Subsequently, genome-wide association studies (GWAS) were conducted for these miRNAs to identify miR-eQTLs in the Rotterdam Study, followed by replication in two independent cohorts [9, 10]. We conducted downstream analyses to elucidate functional characteristics of the findings through cis and trans mapping of miR-eQTLs, cross-phenotype, and multi-omics QTLs analysis and colocalisation. Additionally, a systematic investigation of the effects of genetically determined miRNA levels on a wide range of clinical conditions was conducted using phenome-wide association studies (PheWAS) in the UK Biobank [17, 18] and Mendelian randomisation (MR) to assess causality between miRNAs and clinical conditions [19].

## Results

An overview of the study workflow is presented in Fig. 1. The results described here and the full summary statistics are accessible through the miRNomics atlas (www. mirnomics.com).

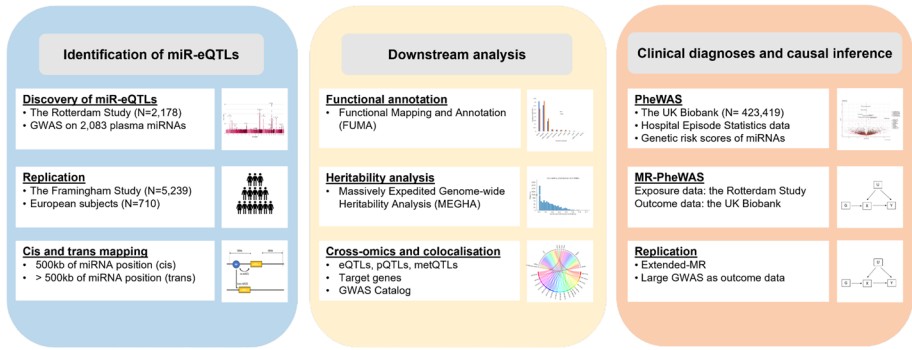

**Fig. 1** An overview of the study workflow

## Genome-wide identification of miR-eQTLs and their functional annotations

### Discovery phase

Plasma levels of 2083 circulating miRNAs were measured (Additional file 1: Table S1) and genome-wide identification of miR-eQTLs was conducted in 2178 participants of the Rotterdam Study (Methods, Additional file 2: Fig. S1). In total, we identified 3292 associations between 1289 SNPs and 63 miRNAs at $P < 2.4 \times 10^{-11}$ (the genome-wide threshold of $P < 5 \times 10^{-08}$ and Bonferroni-correction for 2083 miRNAs) (Additional file 1: Table S3, Fig. 2 and Additional file 2: Fig. S2). The 3292 identified associations included 1733 cis associations (1010 unique SNPs and 32 miRNAs) and 1559 trans associations (294 unique SNPs and 33 miRNAs). Conditional analyses identified 241 conditionally independent associations (113 unique SNPs) at $P < 2.4 \times 10^{-11}$. These included 98 cis associations (57 SNPs and 32 miRNAs) and 143 trans associations (57 SNPs and 32 miRNAs) (Additional file 1: Table S4). The overall proportion of variance explained by each miR-eQTL ranged from 2 to 11%, and 18 miR-eQTLs ($r^2 < 0.6$) were found to explain over 5% of the variation in their corresponding miRNA levels (Additional file 1: Table S5).

### Replication phase

We replicated 1462 associations for 27 miRNAs using the GWAS summary statistics on circulatory miRNA levels from Nikpay et al. [9], (Additional file 2: Fig. S2, Additional file 1: Tables S6–7). The effect estimates demonstrated a strong correlation ($r = 0.82$, $P < 2.2 \times 10^{-16}$) (Additional file 2: Fig. S3). In a secondary analysis, 69% of associations reported by Nikpay et al. [9] and 15% of associations from the Framingham Heart Study [19] were replicated in our study. We finally reported all miR-eQTLs replicated across cohorts, including those that were not initially discovered using a stringent threshold

(See figure on next page.)
**Fig. 2 a** Manhattan plot showing the identified miR-eQTLs. The strongest association for each of 63 miRNAs reaching $P < 2.4 \times 10^{-11}$ is colour-labelled (yellow for cis, and green for trans). Highly pleiotropic loci were identified in locus chr14:100,655,022–101244293, the majority of which were cis-miR-eQTLs and in chr9:136,128,546–13,629,653, the majority of which were trans-miR-eQTLs. This plot only shows associations with $P < 1.0 \times 10^{-5}$. **b** Functional consequences of the identified miR-eQTLs on nearby and far genes. **c** Twenty miRNAs with the highest SNP-based heritability estimates

a.

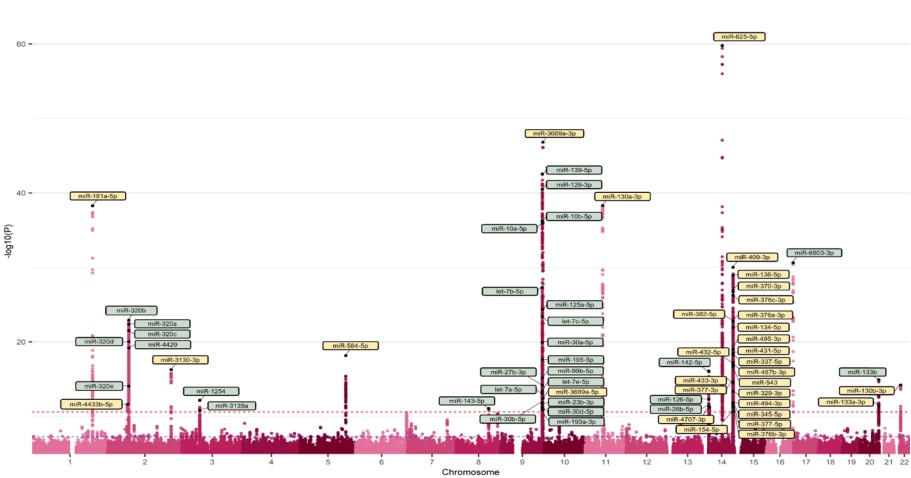

b.

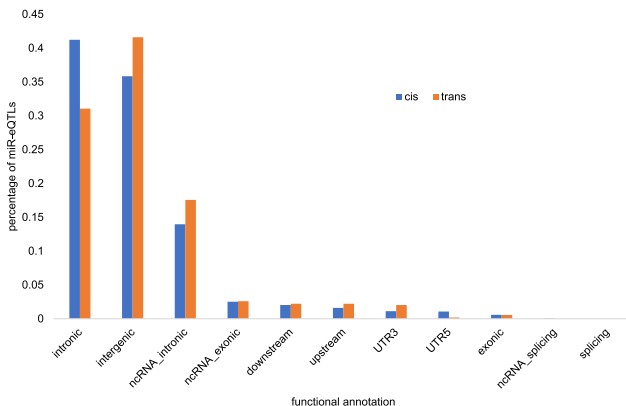

c.

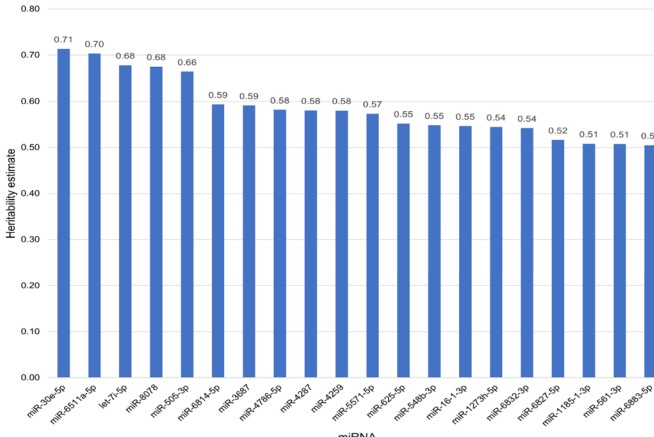

**Fig. 2**　(See legend on previous page.)

($P < 2.4 \times 10^{-11}$) in our discovery. These were considered the most robust findings including 4310 replicated associations for 64 miRNAs (Additional file 1: Table S8). These included associations for 20 miRNAs that originally did not reach our study significance threshold. An example of these were cis-variants for miR-1908-5p, such as rs174561, which was previously reported as miR-eQTL in plasma and other tissues [9, 20].

### Functional annotations

Over 70% of miR-eQTLs were located in the intronic and intergenic regions (Fig. 2, Additional file 1: Table S9). We performed mapping for discovered and replicated miR-eQTLs across studies. These miR-eQTLs were mapped using FUMA (Methods) [21] to identify the genomic loci regulating miRNA expression in plasma. These miR-eQTLs were mapped into 22 genomic loci, of which 11 loci were pleiotropic, i.e., linked to the level of multiple miRNAs (Additional file 1: Table S10). One noteworthy highly pleiotropic locus was identified on chr14:100655022–101244293, known as 14q32 miRNA cluster, regulating 23 miRNAs, predominantly as cis-miR-eQTLs (Additional file 2: Fig. S4). While pairwise phenotypic correlation analysis across all 64 miRNAs with genetic findings resulted in median absolute correlation coefficient of 0.14 (interquartile range (IQR): 0.23), the absolute correlation coefficient between miRNAs in this locus appeared to be higher (median: 0.35, IQR: 0.23) (Additional file 1: Fig. S5). Nevertheless, there remain three miRNAs in this cluster which do not correlate with any other miRNAs (absolute correlation coefficient < 0.3) in the same locus, namely miR-345-5p, miR-411-3p, and miR-433-3p (Additional file 1: Fig. S5). These observations may indicate that one locus could be truly pleiotropic by regulating multiple independent miRNAs.

Another highly pleiotropic locus was on chr9:136,128,546–136,296,530 mapped to *ABO* and other genes (Additional file 1: Table S10) and regulated 18 miRNAs. This locus contained shared trans-miR-eQTLs for several well-known miRNAs, such as miR-10, let-7, and miR-30 families (Additional file 1: Table S10), contributing to 84 out of 143 conditionally independent trans-associations (Additional file 1: Table S4). The median absolute correlation coefficient between these 18 miRNAs was 0.61 (IQR:0.16) (Additional file 1: Fig. S5).

Twelve identified miR-eQTLs located in the miRNA-encoding sequences (seed, mature, or precursor gene of miRNAs) affect the levels of their corresponding miRNA (Additional file 1: Table S11). Forty-two miR-eQTLs were located in the promoter region of miRNAs, including 33 miR-eQTLs in the promoter region of the same miRNAs, and nine in the promoter region for multiple miRNAs (Additional file 1: Table S12). As an example, rs10761364 affected the level of miR-27b-3p in our study and was mapped into the promoter region of a polycistronic miRNA cluster, namely hsa-miR27b, hsa-miR-24–1, and hsa-miR-23b. Several miR-eQTLs for miR-130b-5p were also mapped into the promoter region of miR-301 which belongs to the same family.

### Heritability analysis

The average of heritability estimates for all miRNAs was 0.08 (Additional file 2: Fig. S6). Two miRNAs had a narrow-sense heritability estimate greater than 0.7, namely miR-30e-5p (0.72) and miR-6511a-5p (0.70) (Fig. 2). We found positive correlation between the heritability estimates and the largest proportion of variation in miRNA levels

explained by single miR-eQTL ($r = 0.46$, $P = 3.2 \times 10^{-03}$) (Additional file 2: Fig. S6). We also found that 63 miRNAs with significant findings in our discovery analysis were on average more heritable than the rest of the studied miRNAs as indicated by higher heritability estimates (mean: 0.12 for 63 miRNAs vs mean: 0.08 for 2083 miRNAs). Of these 63 miRNAs, 47 miRNAs were among the well-expressed miRNAs in plasma (Methods), meaning that we could measure these miRNAs reasonably better than the rest.

**Cross-omics and colocalisation analysis**

As miRNAs dictate their role in biological processes by regulating the expression of their target genes, it is interesting to know whether miR-eQTLs are linked to the expression of other genes, including their host and target genes. Combining the identified miR-eQTLs and large-scale blood eQTLs data showed that cis-miR-eQTLs for 39 miRNAs were overlapped with cis-eQTLs for 146 genes (Additional file 1: Table S14), with twelve miRNAs shared cis-miR-eQTLs with their host genes (Additional file 1: Table S15). Colocalisation analysis indicated shared causal variants for four miRNAs and their host genes (PP H4 > 0.7), namely miR-139-3p and *PDE2A*, miR-335-5p and *MEST*, miR-584-5p and *SH3TC2*, and miR-744-5p and *MAP2K4* (Additional file 1: Table S15). We also found an overlap between cis/trans-miR-eQTLs and cis/trans-eQTLs of putative target genes of miRNAs (Additional file 1: Table S16).

We conducted a colocalisation analysis between miRNAs and gene expression across 49 tissues using the GTex dataset [22]. The colocalisation was conducted when cis-miR-eQTLs were found to be at least significantly associated with gene expression ($P < 0.05$) in each tissue. We screened for 64 miRNAs reported in our replication analysis, then tested for 20,979 associations (46 miRNAs and 909 genes across 49 tissues). We identified 450 associations with PP.H4 > 0.7 between 30 miRNAs and 106 genes (Additional file: Table S15). For example, we additionally found evidence of a shared genetic signal between miR-584-5p and its host gene (*SH3TC2*) in the lung and between miR-335-5p and its host gene (*MEST*) in 13 other tissues, including the brain, artery, and adipose tissues. While colocalisation analysis with tissue-level gene expression data allowed us to identify shared genetic signals with host or nearby genes and, to some extent, indicate that the miRNAs expressed in plasma might also be expressed or act in those tissues, it is of importance to do such analysis using tissue-specific miRNA expression when such data is available in a large cohort. This tissue-wide analysis may help to pinpoint the potential source of circulating miRNAs and elucidate their tissue specificity.

Cis-miR-eQTLs of 18 miRNAs were also overlapped with pQTLs for nine proteins. Specifically, the cis-miR-eQTLs for the 14q32 miRNA cluster were shared with pQTLs of *DLK1* located in the nearby genomic region and *SEMG2* in a distant region (Fig. 3, Additional file 1: Table S17). Colocalisation analysis supported shared causal variant for 13 miRNA-protein pairs (all with PP H4 > 0.9), including miR-127-3p, miR-136-5p, miR-431-5p, and miR-433-5p with *DLK1* and *SEMG2*, as well as miR-625-5p with Alpha-(1,6)-fucosyltransferase (Additional file 1: Table S17). Moreover, cis-miR-eQTLs for miR-130a-3p overlapped with pQTLs of Pappalysin-1 (*PAPPA*). Additionally, trans-miR-eQTLs for 11 miRNAs overlapped with pQTLs for 103 proteins (Fig. 3), some of which were target genes of miRNAs, such as miR-126-3p (*TEK*) and miR-145-5p (*MMP1* and *VEGFA*) (Additional file 1: Table S18–S19).

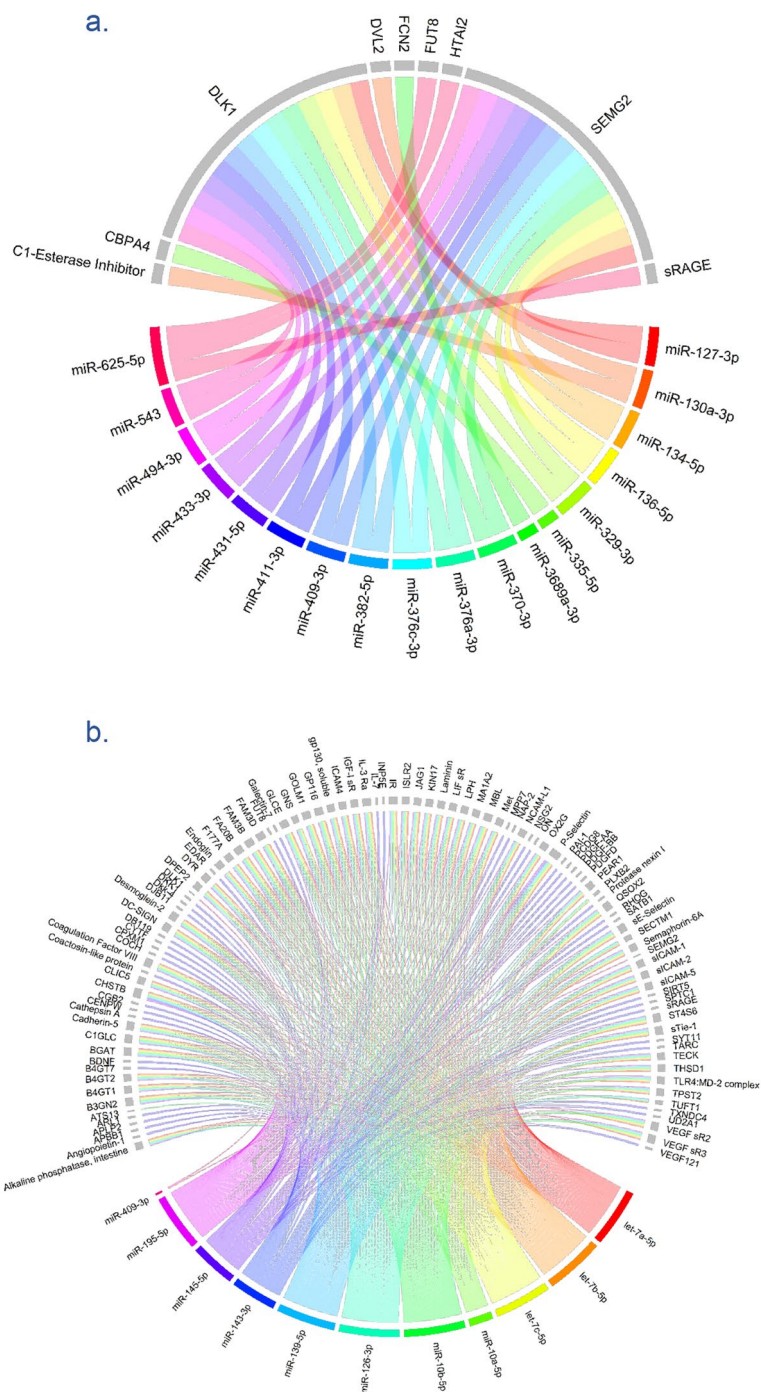

**Fig. 3 a** The overlap between cis miR-eQTLs and proteins (pQTLs). The bottom half of the circle shows miRNAs in different colours, and the top half of the circle (grey coloured) shows the genes. **b** The overlap between trans-miR-eQTLs and proteins (pQTLs). Trans-miR-eQTLs are shown to be more pleiotropic than cis-miR-eQTLs

In our analysis, we found that at least several proteins with shared genetic regulations that their related genes were potential targets for miRNAs, such as miR-127-3p, miR-136-5p, miR-431-5p, and miR-433-5p with *DLK1* and *SEMG2*, as well as miR-625-5p with Alpha- (1,6)-fucosyltransferase (Additional file 1: Table S17). To note,

those miRNA-target pairs were among the predicted miRNA-target interaction (MTI) in TargetScan [5], with none having been validated experimentally in previous studies according to miRTarBase [4] at the time of this analysis. If there is miRNA-target interaction, colocalisation could provide evidence at both gene expression and protein levels, as miRNAs are expected to repress the translation of mRNAs to protein. As an example, miR-625-5p and *FUT8* demonstrated colocalisation with gene eQTLs and pQTLs. Our analysis highlighted the equal importance of in silico and experimental studies to elucidate shared genetic signals underlying potential MTI and validation of those interactions.

We found overlapping cis-miR-eQTLs for miR-1908-5p, miR-148a-3, miR-339-5p, and miR-130a-3p with metabolite-QTLs for 218 metabolites, measured either by the Nightingale or Metabolon platforms. For example, rs174561, located in the precursor gene of miR-1908-5p and intronic to *FADS1*, both known to be associated with lipid and obesity traits, was associated with lipid metabolites. Shared causal variants were identified between miR-1908-5p, miR-148a-3p, and miR-339-5p and lipid metabolites in colocalisation analysis (PP H4 > 0.7) (Additional file 1: Table S20). We also conducted linear regression analysis using individual-level data on metabolite levels to support our genetic findings with metabolite-QTLs. In summary, 20,081 miRNA-metabolite pairs with genetic findings can be tested using individual-level data. Of these, nearly 75% (15,050 pairs) were significant after correcting for multiple testing (FDR < 0.05). These provided further individual-level data analysis that aligned with our findings from publicly available datasets (Additional file: Table S10). Such analysis could not be performed for gene expression and protein abundance due to the lack of data in the Rotterdam Study.

We then examined the association of cis-miR-eQTLs with clinical traits using previous GWAS data and found their associations with mental health, haematological indices, cancers, anthropometric measures, lipid levels, and blood pressure (Additional file 1: Table S21). For example, cis-miR-eQTLs for miR-1908-5p were associated with multiple traits, mainly lipid through *FADS1*, *FADS2*, or *MYRF*, in line with the observed colocalisation of genetic signals with eQTLs and pQTLs. Trans-miR-eQTLs were associated with various diseases, including haematological indices, cardiometabolic, cancer, and allergy.

The trans-regulatory region on Chr.9, mapped to *ABO* gene, was associated with plasma proteins, metabolite levels, and various complex traits mainly of circulatory diseases. For example, rs687289, was associated with the level of 6 miRNAs, is also reported as pQTLs and mQTLs and is associated with GWAS traits such as monocyte count, coagulation factor levels, and pancreatic cancer (Additional file 1: Table S22). This analysis suggests the pleiotropic properties of *ABO* on miRNA expression, by trans-regulating multiple miRNAs in addition to other molecular traits and diseases (Fig. 4).

### Associations of miR-eQTLs with a wide range of clinical diagnoses

To investigate the associations between genetically determined circulating miRNA and a wide range of clinical diagnoses, we conducted a phenome-wide association study (PheWAS) using hospital episode statistics data in 423,419 participants in the UK Biobank (Additional file 1: Fig. S7). We implemented an FDR-based threshold for

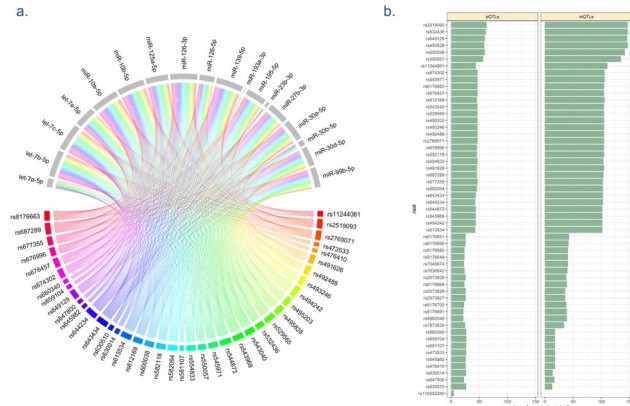

**Fig. 4** The figure indicates pleiotropy of the ABO locus in regulating multiple miRNAs and molecular traits. **a** Genetic variants in the ABO locus (lower half) were found to be associated with 18 miRNAs (upper half). **b** Number of proteins and metabolites associated with each genetic variant in the ABO locus. pQTLs: protein QTLs, mQTLs: metabolite QTLs

every miRNA (in the region spanning 500 kb on either side of the miRNA position) in our PheWAS and MR analyses to enable identifying more instruments and covering more miRNAs. The summary statistics of the FDR-significant cis-miR-eQTLs are provided in Additional file 1: Table S23 and the full results are available through our web tool (www.mirnomics.com).

We identified a single cis-instrument for 85 miRNAs (Additional file 1: Table S23) and multiple cis-instruments for 119 miRNAs. This enabled us to compute genetic risk scores (GRS) for the latter group (Fig. 5). We used the 85 single cis-instruments and 119 miRNA GRS to run PheWAS in the UK Biobank (including 905 phecodes with at least 200 cases across 16 disease groups). Twenty-nine associations were

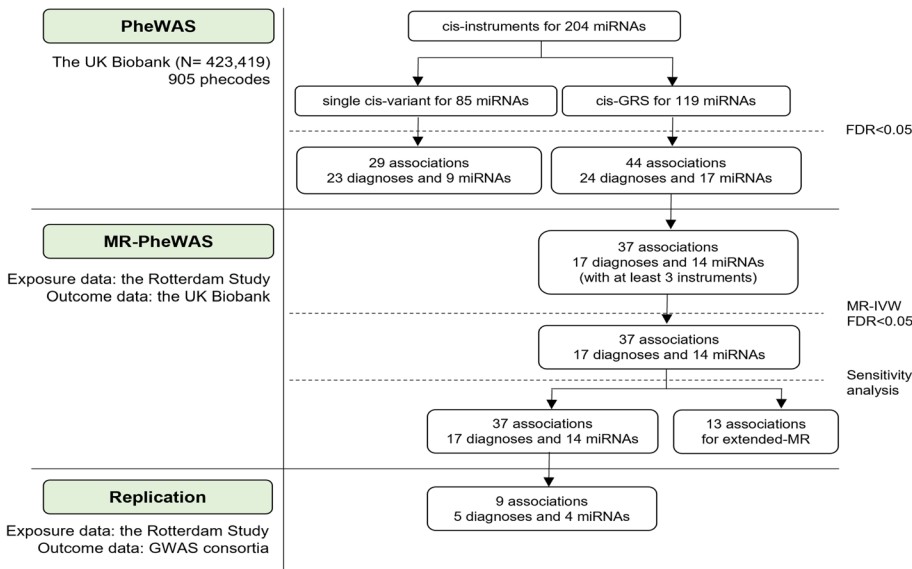

**Fig. 5** The figure depicts a summary of the PheWAS and MR analyses. Cis-variants were used as genetic instruments for miRNAs. When multiple cis-miR-eQTLs were available, cis-GRS was computed for PheWAS. Otherwise, a single variant PheWAS was conducted. MR were conducted for miRNA with at least three instruments. When available, large GWAS data were used to replicate the findings. Otherwise, genome-wide trans-miR-eQTLs were added in the extended-MR

identified in single variant PheWAS, with the strongest association found between rs1254901 (miR-6071) and coronary atherosclerosis (Fig. 6a). Forty-four associations were identified in genetic risk score PheWAS, with the strongest association found between miR-1908-5p and the risk of benign neoplasm of the colon (Fig. 6b). Eleven miRNAs were associated with circulatory disorders (Fig. 6c), with several miRNAs being associated with diagnoses across different disease groups, indicating their pleiotropic properties (Additional file 2: Fig. S8).

MR-PheWAS further identified 37 FDR-significant associations that were robust to sensitivity analyses (Additional file 1: Table S26). Of these, we conducted an extended MR for 13 associations by adding genome-wide significant trans-miR-eQTLs (Methods, Fig. 5), where twelve associations remained significant and were robust to sensitivity analyses (Table 1, Additional file 1: Table S27). For the remaining 24 associations, concordant direction across different MR methods was observed (Additional file 2: Fig. S8).

The associations between miRNAs and obesity-related traits were replicated using the large-scale GWAS data for the outcome, namely between miR-543 and WHR (MR-IVW estimate $= -0.02$, $P = 1.72 \times 10^{-02}$) and between miR-329-3p and BMI (MR-IVW estimate $= -0.03$, $P = 1.89 \times 10^{-02}$) (Table 1, Additional file 2: Fig. S9, Additional file 1: Table S28), with no significant effects in the opposite direction (Additional file 1: Table S29). The observational analysis in the Rotterdam Study ($N = 2740$), adjusting for age, sex, and sub-cohort, also showed a suggestive association in the same direction of effect as in the MR analysis between miR-543 and WHR (estimate $= -0.002$, $P = 0.056$). Through an in-silico search of target genes using TargetScan v7.2 [5] and miRTarBase [4], eighty-two predicted and eighteen validated target genes associated with BMI or WHR were found for miR-543. Likewise, 43 predicted and 58 validated target genes associated with BMI or WHR were identified for miR-329-3p (Additional file 1: Table S30). There was a significant enrichment for BMI or WHR-related genes among validated targets of miR-543 ($P = 9.00 \times 10^{-03}$) and predicted targets of miR-329-3p ($P = 3.18 \times 10^{-02}$).

The most significant association identified in our analysis was the protective effect of miR-1908-5p on the risk of benign neoplasm of the colon (Fig. 6), with no evidence of a causal effect in the opposite direction (Fig. 7). Notably, miR-1908-5p is located in the exonic region of *FADS1* [23]. We found evidence of colocalisation between the expression of miR-1908-5p and the *FADS1* gene in the circulation (PP.H4.abf $= 0.8$), with the most likely candidate causal variant being rs102275 (Additional file: Table S15). The presence of a shared genetic signal between miRNA and host gene raised the question of whether the effect identified between miR-1908-5p and benign neoplasm of the colon has been driven by *FADS1* rather than miR-1908-5p. Colocalisation analysis suggested a shared causal variant between benign neoplasm of colon and miR-1908-5p (PP

(See figure on next page.)

**Fig. 6  a** Enhanced volcano plots for single variant PheWAS. **b** Enhanced volcano plots for GRS PheWAS. The *X*-axis denotes effect estimates for corresponding SNP or GRS. *Y*-axis indicates -log10 of the association *p*-values between each SNP or GRS and clinical condition. Different colours of the dots represent different SNPs. Different shapes show different disease groups. Thresholds of significance are indicated by dashed blue (nominal), red (FDR), and purple (Bonferroni) lines. Plots were only created for SNP and GRS with at least one FDR-significant finding. **c** Number of miRNAs associated with diagnoses in each disease group as identified in PheWAS

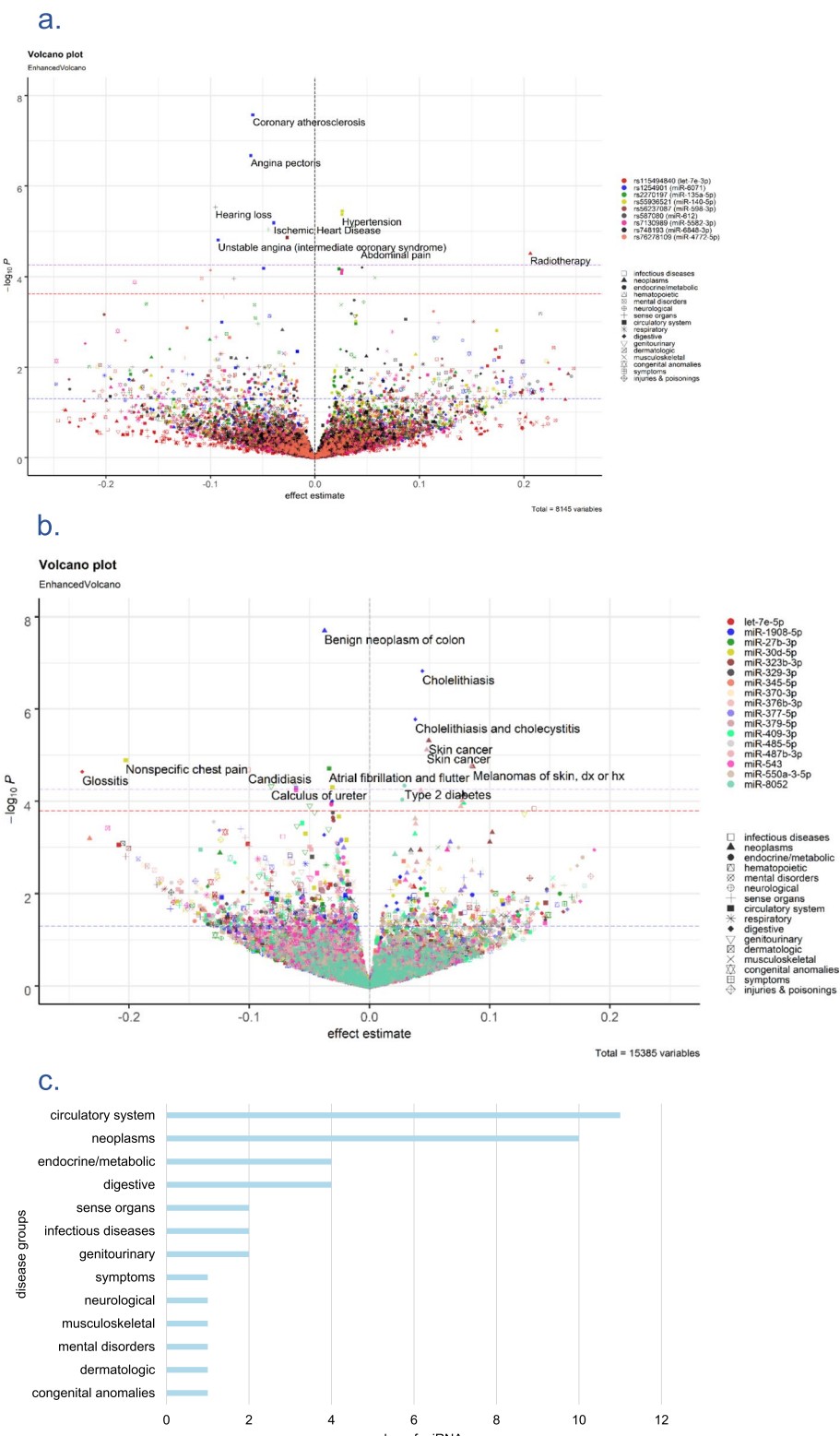

**Fig. 6** (See legend on previous page.)

**Table 1** The results of Mendelian randomisation (MR-IVW) for replicated associations

| Exposure | MR-PheWAS | | | | | | | Validation/replication | | | | | | |
|---|---|---|---|---|---|---|---|---|---|---|---|---|---|---|
| | outcome | n | beta | SE | P | P-het | Egger int P | Outcome | n | beta | SE | P | P-het | Egger int P |
| *Validation in extended-MR* | | | | | | | | | | | | | | |
| miR-30d-5p | Angina pectoris | 5 | −0.47 | 0.10 | $1.5 \times 10^{-6}$ | $3.4 \times 10^{-1}$ | $2.9 \times 10^{-1}$ | Angina pectoris | 6 | −0.34 | 0.12 | $6.0 \times 10^{-3}$ | $3.1 \times 10^{-2}$ | $2.7 \times 10^{-1}$ |
| | Coronary atherosclerosis | 5 | −0.39 | 0.09 | $7.6 \times 10^{-6}$ | $3.6 \times 10^{-1}$ | $2.5 \times 10^{-1}$ | Coronary atherosclerosis | 6 | −0.26 | 0.11 | $1.8 \times 10^{-2}$ | $2.8 \times 10^{-2}$ | $2.4 \times 10^{-1}$ |
| | Nephrotic syndrome | 5 | 1.91 | 0.41 | $3.9 \times 10^{-6}$ | $6.1 \times 10^{-1}$ | $5.7 \times 10^{-1}$ | Nephrotic syndrome | 6 | 1.32 | 0.50 | $8.9 \times 10^{-3}$ | $7.0 \times 10^{-2}$ | $9.9 \times 10^{-1}$ |
| | Nonspecific chest pain | 5 | −3.07 | 0.56 | $4.9 \times 10^{-8}$ | $7.9 \times 10^{-1}$ | $9.2 \times 10^{-1}$ | Nonspecific chest pain | 6 | −2.03 | 0.82 | $1.4 \times 10^{-2}$ | $1.2 \times 10^{-2}$ | $6.5 \times 10^{-1}$ |
| miR-323b-3p | Melanomas of skin | 6 | 0.45 | 0.08 | $1.4 \times 10^{-7}$ | $8.8 \times 10^{-1}$ | $5.7 \times 10^{-1}$ | Melanomas of skin | 7 | 0.31 | 0.11 | $4.7 \times 10^{-3}$ | $4.3 \times 10^{-2}$ | $8.1 \times 10^{-1}$ |
| | Obesity | 6 | −0.16 | 0.03 | $6.6 \times 10^{-6}$ | $7.7 \times 10^{-1}$ | $7.1 \times 10^{-1}$ | Obesity | 7 | −0.11 | 0.05 | $1.9 \times 10^{-2}$ | $4.6 \times 10^{-2}$ | $8.7 \times 10^{-1}$ |
| | Overweight or obesity | 6 | −0.15 | 0.03 | $8.5 \times 10^{-7}$ | $7.1 \times 10^{-1}$ | $6.9 \times 10^{-1}$ | Overweight or obesity | 7 | −0.11 | 0.04 | $1.7 \times 10^{-2}$ | $5.6 \times 10^{-2}$ | $8.6 \times 10^{-1}$ |
| | Skin cancer | 6 | 0.26 | 0.05 | $2.2 \times 10^{-7}$ | $2.9 \times 10^{-1}$ | $9.6 \times 10^{-1}$ | Skin cancer | 7 | 0.21 | 0.06 | $3.4 \times 10^{-4}$ | $5.2 \times 10^{-2}$ | $9.0 \times 10^{-1}$ |
| | Viral enteritis | 6 | 0.72 | 0.15 | $1.7 \times 10^{-6}$ | $8.7 \times 10^{-1}$ | $7.3 \times 10^{-1}$ | Viral enteritis | 7 | 0.61 | 0.13 | $3.2 \times 10^{-6}$ | $6.7 \times 10^{-1}$ | $6.6 \times 10^{-1}$ |
| miR-409-3p | Melanomas of skin | 12 | 0.25 | 0.05 | $7.7 \times 10^{-6}$ | $1.1 \times 10^{-1}$ | $5.0 \times 10^{-1}$ | Melanomas of skin | 13 | 0.24 | 0.05 | $3.3 \times 10^{-6}$ | $1.4 \times 10^{-1}$ | $4.7 \times 10^{-1}$ |
| *Replication using large GWAS summary statistics* | | | | | | | | | | | | | | |
| miR-329-3p | Overweight or obesity | 10 | −0.15 | 0.03 | $1.5 \times 10^{-7}$ | $3.2 \times 10^{-1}$ | $2.4 \times 10^{-1}$ | BMI | 6 | −0.03 | 0.01 | $1.9 \times 10^{-2}$ | $4.9 \times 10^{-2}$ | $9.9 \times 10^{-2}$ |
| | Obesity | 10 | −0.15 | 0.03 | $3.9 \times 10^{-8}$ | $3.5 \times 10^{-1}$ | $2.4 \times 10^{-1}$ | | | | | | | |
| miR-543 | Overweight or obesity | 8 | −0.15 | 0.03 | $9.5 \times 10^{-9}$ | $4.7 \times 10^{-1}$ | $8.4 \times 10^{-1}$ | WHR | 7 | −0.02 | 0.01 | $1.7 \times 10^{-2}$ | $7.7 \times 10^{-1}$ | $9.0 \times 10^{-1}$ |
| | Obesity | 8 | −0.15 | 0.03 | $1.1 \times 10^{-8}$ | $4.9 \times 10^{-1}$ | $8.9 \times 10^{-1}$ | | | | | | | |

*BMI* Body mass index, *WHR* Waist-to-hip ratio, *n* is the number of genetic instruments used in the analysis, *SE* Standard error, *P*-het denotes *P*-value for heterogeneity of MR-IVW estimates, *Egger int P*, *P*-values for MR Egger intercept. The summary statistics presented are based on MR-IVW. Full results for other MR methods are presented in Additional file 1: Tables S27 and S28. In extended MR, trans-miR-eQTLs were added to the cis-miR-eQTLs as genetic instruments for miRNAs

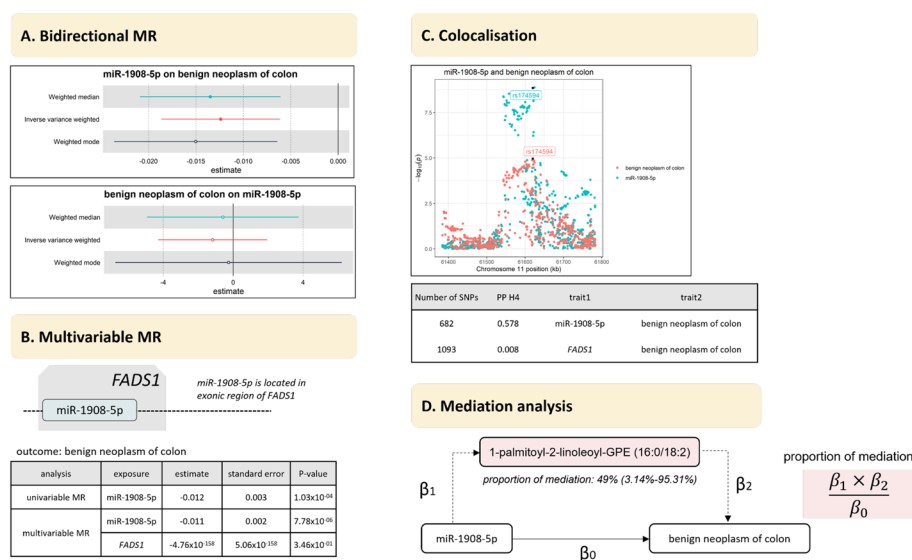

**Fig. 7** The figure shows our investigation for the causal association of miR-1908-5p with colon cancer and the potential mediators. **A** Bidirectional MR analysis showing the potential causal effect of miR-1908-5p on the risk of benign neoplasm of the colon, with no effect observed in the opposite direction. **B** The effect of miR-1908-5p on the risk of benign neoplasm of the colon remained significant when adjusting for the genetic effect of the host gene, as evidenced in multivariable MR. **C** Pairwise colocalisation analysis showed evidence of a shared causal variant between miR-1908-5p and benign neoplasm of the colon (PP H4 = 0.57). **D** Mediation analysis conducted using two-step MR estimated 49% of the total effect of miR-1908-5p on the disease is mediated by 1-palmitoyl-2-linoleoyl-GPE (16:0/18:2)

H4 = 0.58), but not for *FADS1* (PP H4 = 0.008) (Fig. 7). Our multivariable MR, where we treated both miR-1908-5p and *FADS1* as exposures, showed an attenuation of the effect of miR-1908-5p (effect estimate = $-0.010$, $P = 7.78 \times 10^{-06}$), but no effect of *FADS1* ($P = 0.67$). Overall, these suggested that miR-1908-5p was the putative causal factor that altered disease risk, with the effect likely being independent of the host gene.

To elucidate pathways mediating the effect of miR-1908-5p on benign neoplasm of the colon, we conducted further MR analysis looking at the effect of miR-1908-5p on metabolite levels using the Metabolon platform [24], which annotated and measured metabolite levels across different classes. This analysis identified 102 metabolites affected by miR-1908-5p (Additional file 1: Table S31, Additional file 2: Fig. S10), of which 12 metabolites were also found to affect the risk of disease—all belonging to lipid class (Additional file 1: Table S32, Additional file 2: Fig. S10). Multivariable MR showed 1-palmitoyl-2-linoleoyl-GPE (16:0/18:2) remaining as the only significant metabolite ($P = 4.67 \times 10^{-03}$) (Additional file 2: Fig. S10), in line with the results of MR-BMA (Additional file 1: Table S33). Finally, mediation analysis showed that 1-palmitoyl-2-linoleoyl-GPE (16:0/18:2) mediated 49% (3.14–95.31%) of the total effect of miR-1908-5p (Additional file 1: Table S34). Further analysis showed no significant effect of miR-1908-5p when adjusting for the 12 candidate metabolites, indicating that the overall effect of miRNA may drive through all those metabolites (Additional file 1: Table S35). The association with lipids found here aligns with our previous study reporting an association between miR-1908-5p with LDL-cholesterol, total cholesterol, triglyceride, and HDL-cholesterol [25].

## Discussion

We present a comprehensive study of genetic regulation and disease associations of plasma circulating miRNAs using population-level data. The study is currently the most extensive single-site analysis of over 2000 circulating miRNAs in 2178 individuals from the Rotterdam Study cohort, followed by replication in two independent cohorts. Our study has expanded the coverage of miRNAs compared to the previous study covering 280 circulatory miRNAs in whole blood in the Framingham Heart Study [10]. Moreover, the sample size in our study is three times larger than the previous study by Nikpay et al. [9], which measured 2083 circulatory miRNA levels using the same miRNA profiling method.

We found significant associations between 1289 SNPs and 63 miRNAs. Our cross-omics QTLs and colocalisation analyses showed that circulating miRNAs could be good proxies for the activity of miRNAs in target tissues which regulate plasma levels of genes, proteins, and metabolites. We revealed the consequences of alteration in plasma miRNA levels on a wide range of clinical conditions, where the causal and pleiotropic effects of identified miRNAs were also investigated in the UK Biobank. Finally, we were able to highlight target genes and pathways regulated by some of the identified miRNAs in the context of their associated clinical conditions.

We identified 3292 genetic associations of common variants that may control the plasma levels of 63 miRNAs. We replicated 65% of these associations in independent cohorts, including trans-miR-eQTLs whose replication was previously minimal. Our results showed 63 out of the 2083 studied miRNAs (approximately 3% of all miRNAs and 10% of the well-expressed miRNAs) have common variants associated with their plasma levels, which could be a relatively small proportion compared to those identified for messenger RNA eQTLs. Although sample size remains a limitation, this may imply that miRNAs have stronger selective constraints that limit their variability [26], such that common variants do not show strong effects. Our heritability analysis revealed the modest effect of genetic variants on plasma miRNA levels, as also shown by the small variation explained by miR-eQTLs, which could act as a mechanism to maintain biological function during evolution. Nevertheless, the positive correlation between the heritability estimates and the largest proportion of variation in miRNA levels explained by single miR-eQTL indicated that higher heritability corresponds with more pronounced regulation by the genetic components.

Our integrative QTL analysis showed that miR-eQTLs colocalise with gene expression and protein QTLs of their target genes, supporting the role of miRNAs in gene regulation and translational repression. Since some target genes tend to be clustered to miRNAs according to their function [27, 28], these shared miR-eQTLs might have biological relevance. Cis-miR-eQTLs that overlap with trans-mRNA-eQTLs might point to the downstream regulatory effect from miRNAs to their (direct or indirect) target genes. On the other hand, for cis-mRNA-eQTLs overlap with trans-miR-eQTLs, the effect might be going from the genes to miRNAs, pointing to bidirectional interaction between miRNAs and target genes as a feedback mechanism [29, 30]. However, when trans-miR-eQTL overlap with trans-mRNA-eQTLs without evidence of miRNA and target gene interaction, a third factor, such as upstream regulatory mechanism, may have contributed to simultaneous changes in miRNA and gene expression. As an example, a genetic variant

could affect the regulatory region shared between miRNA and a gene that is co-expressed. It can be hypothesised that cis- and trans-miR-eQTLs might have different clinical relevance. The magnitude of associations between miRNAs and complex traits appeared closer to the null when trans-miR-eQTLs were added as instruments in our study. Trans-miR-eQTLs might affect the stability of mature miRNAs, whereas cis-miR-eQTLs influence the hairpin structure and regulate the expression of primary miRNAs [9].

Given that each miRNA potentially regulates multiple target genes and pathways [1, 2, 5], even small changes in miRNA expression may result in considerable consequences. This concept aligns with the strong evolutionary constraint on miRNAs and their binding sites in gene 3-UTRs in humans and other species [5]. Moreover, the seed, mature, and precursor regions of miRNA genes are known to have a lower density of genetic variation than the whole genome [31]. Our study shows that the genetic variants in those regions could have functional importance, such as affecting miRNA transcription. Examples were shown in previous studies, such as by Toste et al. reporting miR-eQTLs of miR-1908-5p, miR-4707-3p, and miR-323b-3p that were predicted to alter the corresponding pri-miRNA hairpin secondary structure [20]. Our previous work has also shown that variants residing in miRNA-related sequences have functional relevance [32]. This functional consequence occurs by interfering with the processing of precursor to mature miRNA or the interaction between mature miRNA and target genes, resulting in gain and loss of function, which could deregulate biological pathways [33, 34].

Human miRNAs can be categorised into families with similar functions due to their conserved structures in the mature or seed sequences [35] and clusters when they are encoded from the same region in our genome [3]. Here, our results showed that the 14q32 miRNA cluster shares cis-regulatory variants. We also showed that multiple miRNAs are regulated by shared miR-eQTLs [36], such as the pleiotropic trans-miR-eQTLs in the *ABO* gene. This finding agrees with the concept that miRNAs can work in networks to control gene expression and pathways underlying diseases [37].

The pleiotropic loci identified in our study are associated with multiple miRNAs, some of which are phenotypically (their expression levels) correlated to each other. Both genetics and environmental factors influence human phenotypes and the correlation between them [38]. Phenotypic correlation could arise due to several reasons, such as shared genetic and environmental determinants or the presence of causal relationships between phenotypes. While genome-wide genetic correlation could be similar to phenotypic correlation in many instances, the genetic contribution on a locus basis could be different [39]. We acknowledge that one needs to be careful when looking at the pleiotropic effect of miRNAs. Future studies could try to disentangle the true pleiotropic effect from the phenotypic correlation between miRNAs or rather to take advantage of these correlations to improve power in genetic discovery.

The pleiotropy of the *ABO* locus has been reported previously by being associated with haematological traits across different populations [40, 41], immune-related proteins and metabolites [42], and cardiovascular traits [43]. Several families sharing trans-regulatory variants in *ABO*, such as the miR-10 family, miR-30 family, let-7 family, and miR-139-5p, were well-known in cardiometabolic traits [44–46]. Endothelial miR-10a/b showed low expression in regions susceptible to atherosclerosis accompanied by up-regulation of Homeobox A1 (*HOXA1*), an experimentally validated target of miR-10a

[47, 48]. The association of miR-10bA with lipid traits was also reported through trans-regulatory in the *ABO* locus [9]. Overexpression of let-7 g in mice resulted in impaired glucose tolerance [49], and knockdown of the let-7 family improved glucose tolerance in mice [50]. The plasma level of miR-139-5p is associated with type 2 diabetes [51], and their up-regulation was found in the peripheral blood of hyperglycaemia patients through suppression of *FoxO1* and *FoxP1* [52]. Our findings further strengthen the relevance of miR-eQTLs in *ABO* to cardiovascular traits acting through trans-regulatory mechanisms.

We used an FDR-based threshold for every miRNA ($\pm 500$ kb of the miRNA position) in the PheWAS and MR analysis to enable identifying more instruments and covering more miRNAs. This decision was made based on several reasons: (1) cis miR-eQTLs are considered biologically relevant, and FDR-based methods are commonly used in cis-eQTL discovery [53]; (2) the main concern with a relaxed *P*-value threshold in the MR analysis is the possibility for weak instruments bias, whereas such bias tends to be towards the null (false negative) in the setting of two-sample MR, as implemented in our study; (3) we replicated our PheWAS and MR analysis using independent datasets to avoid false positive findings.

Several associations with complex traits highlighted in this study were reported in the literature. For example, miR-543 was released in plasma following a high-fat diet [54], which could be a physiological response to reduce the risk of obesity. Target genes of miR-329-3p were involved in lipid and glucose metabolism in rats [55]. Low miR-329 expression was observed in melanoma cells, while miR-329 mimics could suppress the progression of melanoma [56]. The effect in tumour tissue for miR-329 and miR-1908-5p [14, 56] was opposite compared to our MR analysis which better captures the lifetime effect of miRNAs. This suggests the changes in the level of miRNAs in tumour tissue might be the consequence of disease processes and supports the hypothesis that the dysregulation of miRNA in diseased tissue might arise from negative feedback by downstream genes [29, 30]. It is also possible that the genetic effects have been buffered by canalisation [19], where people with a genetically higher level of miRNAs since the intrauterine period might be resistant to the effect of higher miRNAs throughout life.

We and others have previously reported the association of miR-1908-5p and lipid traits [25, 57, 58], anthropometric traits and cancer [14]. Our current and previous studies both highlight the relevance of lipid pathways for miR-1908-5p. Furthermore, here we show how this pathway links miR-1908-5p and benign neoplasm of the colon. While it is known that co-expression of miRNAs and their host genes could occur through modification of promoter activity, chromatin accessibility, transcription factor binding, or DNA methylation [10], many miRNAs also have their own promoters [59].

Our colocalisation and multivariable MR analysis indicated that the genetic effects of miRNAs on complex disorders could be independent of the host genes, as previously reported [10, 60]. In this study, we showed the importance of disentangling the effect of miRNA from host genes. There are several approaches that can be used to fulfil this. If both miRNA and host gene expression data is available in the same participants, conditional analysis could be performed with adjustment on the expression of host genes for any associations identified for miRNAs, as demonstrated previously [10]. If the data is not available for the same participants, genetic association data could be used within a

multivariable framework, such as through multivariable MR analysis. Our example for miR-1908-5p, *FADS1*, and benign neoplasm of colon serves as an example of the latter. This approach should be carried out in future research to exclude the possibility of the host gene being the key player rather than the miRNA of interest, given that both often have shared genetic regulation.

Our colocalisation analyses showed shared genetic signals between miRNAs and their host or target genes, proteins, and metabolites. We provide another layer of evidence for a correlation between miRNA and metabolite abundance using individual-level data in the Rotterdam Study. Similarly, this analysis could also be done with their respective host or target genes abundance. However, gene expression data is not available for the same participants in the Rotterdam Study at the time of the analysis.

We should underline several aspects to be considered when attempting to replicate miR-eQTLs across studies. First, we found that fewer trans- were replicated than cis-miR-eQTLs, as observed in the large eQTL analysis as well [53]. At the genome-wide significance threshold, we observed lower replication rates for miR-eQTLs, with an overall decrease of 22.6%. This decline was much more pronounced in the replication of trans (32.2%) compared to cis-miR-eQTLs (9.2%). This indicates that many trans-signals detected at the conventional genome-wide threshold contained non-genuine signals that were less replicable, consistent with a previous study [10]. Trans-eQTLs are known to have weaker effects, be less replicable, and be more tissue-specific [61–63] than cis-eQTLs. Trans-miR-eQTLs were found to be more pleiotropic by being associated with other omics QTLs. This made them unsuitable for instrumenting miRNAs due to the risk of horizontal pleiotropy which could violate the MR assumptions. It is therefore recommended to use cis miR-eQTLs, although caution remains needed, given that they are also often shared with host or nearby genes, as shown in our example on miR-1908-5p and *FADS1.*

Second, the concordant direction with those reported by Nikpay et al. [9] suggested that the type of biological sample and profiling method could have an effect. The lower replication rate in the Framingham Heart Study is likely due to differences in the type of sample (whole blood vs plasma), as previously reported [64], and the miRNA profiling method (qPCR vs targeted RNA-seq). Third, one should consider any systematic difference in participants' characteristics across studies. This study came from a population-based cohort which makes the findings more generalisable. Other studies were in obese individuals [9] or enriched for a specific disease [11], making it particularly useful for investigating the relevant disease but not for a wide range of complex traits and disorders. Finally, since the proportion of variation explained by some miR-eQTLs is relatively small, larger GWAS meta-analyses will be warranted to identify more miR-eQTLs. In particular, incorporating diverse ancestries could generate more transferrable findings for a wider population.

## Conclusions

Collectively, the integration of genomics, other omics, and clinical data at the population level in this study has provided a better understanding of the genetic regulation of miR-NAs and the impact of perturbations of plasma levels of miRNAs on a wide range of clinical traits. Although it is unlikely a single miRNA or its target genes will be entirely responsible

for causing a disease, it is plausible that the effect of identified miRNAs to be mediated at least in part through its target genes implicated in the disease mechanisms. Our approach allows generating testable hypotheses for further functional and clinical studies to dissect the underlying molecular mechanisms and cellular pathways of various traits and diseases.

We have generated a web-based tool: miRNomics atlas (www.mirnomics.com) to release the results of our study publicly available. This tool allows the use of genetic association data of miR-eQTLs, serving as valuable resources for future research to decipher the association and causal role of miRNAs in human diseases and their regulatory pathways.

## Methods

### Cohort description

The Rotterdam Study (RS) is a large prospective population-based cohort study among middle-aged and elderly in the suburb Ommoord in Rotterdam, the Netherlands. In 1990, 7983 inhabitants aged 55 years old and older were recruited to participate in the first cohort (RS-I). In 2000, the study was extended with a second cohort of 3011 participants (RS-II) who became 55 years old or moved into the study district since the beginning of the study. In 2006, a further extension of the cohort (RS-III) was initiated, including 3932 participants aged 45–54 years. In 2016, the recruitment of another extension started (RS-IV), targeting participants aged 40 years and over, adding 3005 new participants. Data on diverse clinical outcomes are collected through follow-up visits every 3–5 years. A detailed description of the Rotterdam Study can be found elsewhere [65].

### Circulating miRNA levels

Plasma cell-free miRNA levels were determined using the HTG EdgeSeq miRNA Whole Transcriptome Assay (WTA) to quantitatively detect the expression of 2083 human miRNA transcripts (Additional file 1: Table S1) (HTG Molecular Diagnostics, Tuscon, AZ, USA) and using the Illumina NextSeq 500 sequencer (Illumina, San Diego, CA, USA). This method characterises miRNA expression patterns and measures the expression of 13 housekeeping genes to allow flexibility during data normalisation and analysis. HTG EdgeSeq has included only high-confidence miRNAs according to their in-house pipeline. Quantification of miRNA expression was based on counts per million (CPM). Log2 transformation of CPM was used as standardisation and adjustment for the total reads within each sample. MiRNAs with log2 CPM 50% values above the lower limit of quantification (LLOQ). Out of 2083 miRNAs, 591 were well-expressed in the samples (Additional file: Table S1).

Log2 transformation of CPM was used as standardisation and adjustment for total reads within each sample. More information on the procedure is presented in Additional file 2: Methods S1.

### Population for analysis

The miRNA expression profiling was performed for 2754 participants randomly selected from three sub-cohorts (RS-I-4, RS-II-2, and RS-IV-1) in the Rotterdam Study [65].

Genotype data were available for 2,435 of the participants. After excluding participants of non-European ancestries and relatives based on kinship coefficient > 0.088 were excluded, 2178 participants were included in the analysis (Additional file 2: Fig. S1). The clinical characteristics of participants are summarised in Additional file 1: Table S2.

For a subset of participants, circulatory miRNAs and metabolite abundance measures were measured in the same subjects of the Rotterdam Study. We used this individual-level data to perform a linear regression analysis to support our genetic findings. However, at the time of our analysis, there was no overlap between miRNA and gene expression or protein abundance in the Rotterdam Study to assess their correlation.

### Identification and mapping of miRNA expression quantitative trait loci

Blood samples were drawn at baseline and genotyping was performed using the Human-Hap550 Duo BeadChip (Illumina, San Diego, California) for RS-I and RS-II and the Global Screening Array (GSAMD-v3) Illumina array for RS-IV. Quality control and imputation steps for the genetic data are available in Additional file 2: Methods S2.

Identification of genetic variants associated with miRNA expression in plasma, or so-called miRNA expression quantitative trait loci (miR-eQTLs), both acting in proximity (cis) or distant (trans), was performed through genome-wide association studies (GWAS) for each of 2083 miRNAs. Given the high number of miRNAs, GWAS was performed within the high-dimensional analysis framework (HASE) to reduce the computational burden and enable efficient implementation of GWAS on thousands of phenotypes [66]. Multiple linear regression was used to test for association between genetic variants and plasma miRNA level, with miRNA level as the outcome and expected genotypes count from imputation as predictors, with adjustment for age, sex, sub-cohort, and the first five principal components to account for population stratification.

We used the genome-wide threshold of $P < 5 \times 10^{-08}$ and Bonferroni-corrected for 2083 miRNAs ($P < 2.4 \times 10^{-11}$) to identify significant associations. Associations reaching the significance threshold in the Rotterdam Study were taken forward for replication in a published miR-eQTLs study by Nikpay et al. [9]. Similarly, associations identified in previous GWAS by Nikpay et al. [9] and Huan et al. in the Framingham Heart Study [10] were also tested for replication. We harmonised the alleles so that the effect estimates between discovery and replication cohorts corresponded to the same effect alleles. Replication was defined when the associations between SNP and miRNA were Bonferroni-significant in an independent cohort with a concordant direction of effect. Further description on the replication is available in Additional file 2: Methods S3.

### Conditional analysis

We conducted multi-SNP-based conditional and joint association analysis implemented in GCTA-COJO [67] to identify conditionally independent association signals within 1MB region of the lead SNPs. In brief, this analysis performed stepwise selection to select SNPs based on conditional $P$-values and provided joint effects of selected SNPs after the model has been optimised. We used genetic data from RS-I ($N = 6291$) of European ancestries to compute the LD reference panel and applied the following filters: minor allele frequency (MAF) > 0.05, conditional $p$-value $2.4 \times 10^{-11}$, collinearity threshold of 0.9, and the assessment window of 10,000 bp.

### Functional annotation of miR-eQTLs

SNPs located $\pm 500$ kb upstream and downstream of the start position of mature miR-NAs were identified as cis, and those located more than $\pm 500$ kb away were identified as trans. The web-based tool Functional Mapping and Annotation (FUMA) was used to annotate miR-eQTLs [21]. We also checked whether miR-eQTLs are in the promoter region of the primary miRNA transcripts as annotated by FANTOM [68].

We used the SNP2GENE process in FUMA [21] to annotate miR-eQTLs into genomic risk loci and mapped them to genes according to their position. Independent significant miR-eQTLs were defined as those with $P < 2.4 \times 10^{-11}$ in the discovery or those replicated in independent cohorts and in moderate LD with each other ($r^2 < 0.6$). LD calculation was referenced based on 1000 Genomes phase 3 panel. These SNPs were further clumped to lead SNPs ($r^2 < 0.1$). Genomic risk loci were then defined based on the lead SNPs when they overlapped with a maximum distance of 250 kb between LD blocks. Details of the functional annotation are provided in Additional file 2: Methods S4.

### Heritability analysis

The SNP-based heritability estimates for 2083 circulating miRNAs were obtained using massively expedited genome-wide heritability analysis (MEGHA) [69]. A genetic relationship matrix was constructed from 1000 Genome imputed genotypes filtered on imputation quality ($< 0.5$) and allele frequency ($< 0.1$) using GCTA [70]. After applying a stringent cut-off of 0.025 for genetic relatedness, 1506 individuals were used for heritability estimation. Using MEGHA, the genetic relationship matrix, and age and sex as covariates, we computed the heritability and uncertainty *p*-values based on 1000 permutations.

### Cross-omics and colocalisation analysis

As miRNAs dictate their role in biological processes by regulating the expression of their target genes, it is interesting to know whether miR-eQTLs are linked to the expression of other genes, including their host and target genes. To explore this, we sought overlaps between replicated miR-eQTLs and gene expression (eQTLs) in whole blood (eQTL-Gen) [53] and across 49 tissues (GTex v8) [22], protein (pQTLs) [71–75], and metabolite-QTLs (mQTLs) [71–73] (Additional file 2: Methods S5). We further checked if any of the genes or proteins that shared QTLs were predicted as target genes of miRNAs in miRNA target prediction databases (TargetScan v7.2 [5] and miRTarBase [4]). Colocalisation analysis was also conducted when there was overlap between cis-miR-eQTLs and other omics (eQTLs/pQTLs/mQTLs) using Bayesian framework to test for the presence of shared causal variant [76] (Additional file 2: Methods S5).

### Linear regression analysis

This analysis was performed in the RS-I-4 cohort of the Rotterdam Study to investigate the relationship between miRNA expression levels and metabolite concentrations. Metabolites from the Metabolon platform (1087 metabolites, 512 participants) and the Nightingale platform (249 metabolites, 975 participants) were used. The analysis involved performing linear regression for each miRNA-metabolite pair while adjusting for sex, age, smoking status, BMI, red blood cell count, white blood cell count, plate

number and well location related to miRNA measurement. The model was fitted using the lm function in R, with adjustment to control for the false discovery rate (FDR) using the Benjamini–Hochberg method [77].

### Phenome-wide association studies

Phenome-wide association study (PheWAS) was conducted using hospital episode statistics data in 423,419 participants in the UK Biobank to investigate associations between genetically predicted circulating miRNA and a wide range of clinical diagnoses (Additional file 2: Fig. S3, Additional file 2: Methods S6). We used independent cis instruments in the primary analysis, with trans instruments added in the sensitivity analysis. Genetic risk score (GRS) was computed for every miRNA when multiple independent instruments were present, otherwise, a single variant was used (Additional file 2: Methods S6). ICD (ninth and tenth editions) codes from the hospital episode statistics data in the UK Biobank were aligned into phecodes to identify clinically related phenotypes. The analysis was limited to phecodes with at least 200 cases to allow sufficient power for MR analysis [78]. PheWAS was conducted using the PheWAS package in R [79]. The false discovery rate (FDR) was calculated for each miRNA-GRS to account for multiple testing [77].

### Mendelian randomisation

We conducted a two-sample Mendelian randomisation (MR) analysis to assess the causal relationship between candidate miRNAs and clinical diagnoses or other omics layers. For clinical diagnoses, we implemented MR in PheWAS analysis (MR-PheWAS). Analysis was conducted when miRNAs had at least three or more independent instruments to perform robust MR methods. Details on our MR analysis can be found in Additional file 2: Methods S7. The multiplicative random effect inverse variance weighted method (IVW) was used in the main analysis to combine the effect estimates of the genetic instruments assuming all instruments are valid [80]. Robust MR methods which allow the inclusion of pleiotropic variants were used as a sensitivity analysis, including weighted median (WM) or MR-Egger [81–83]. The agreement among different MR methods was examined to support a robust estimation of causal effects. Since a liberal LD threshold ($r^2 < 0.1$) was used for clumping, a further sensitivity analysis was conducted by incorporating the correlation matrix between genetic instruments in the fixed effect IVW method [84]. MRPRESSO was used to detect outliers [85] and MR analysis was repeated after excluding outliers. We used multivariable MR to rule out the effect of other factors, run mediation analysis to disentangle the effect of miRNA and host gene on the disease risk, or to identify potential metabolites mediating the effect of miRNA. When assessing potential mediators linking miRNA and disease, both classic MVMR and Bayesian (MR-BMA) methods [86] were used in complementary. When a mediator was identified, a two-step MR was conducted to assess the proportion of mediation [87]. Predicted and validated target genes of disease-associated miRNAs were retrieved, and enrichment analyses were conducted as described in our previous work [27].

## Supplementary Information

### Acknowledgements

We would like to thank all participants of the Rotterdam Study and the UK Biobank. This work was enabled by the computing resources and support from the Imperial College Research Computing Service and Erasmus MC. We thank Loukas Zagkos for helping with the visualisation of the results, and Devendra Meena and Georg Otto for technical support.

### Review history

The review history is available as Additional file 3.

### Peer review information

### Authors' contributions

MG and AD designed the study and oversaw the research. RM did the statistical analyses and wrote the first draft of the manuscript. MMM and AvH helped with sub-analyses. MG, MAI, and JVM provided resources and data. TH and DL helped with the replication of the study in the FHS. All authors interpreted the data and commented on the draft report. All authors approved the final manuscript.

### Funding

This project is supported by the Erasmus MC Fellowship (EMCF20213) of MG. RM is supported by the President's PhD Scholarship from Imperial College London. AD is funded by a Wellcome Trust seed award (206046/Z/17/Z). PE acknowledges support from the Medical Research Council (MR/S019669/1) for the MRC Centre for Environment and Health, the British Heart Foundation (RE/18/4/34215) for the Imperial BHF Centre for Research Excellence, the UK Dementia Research Institute (MC_PC_17114) and the National Institute for Health Research Imperial College Biomedical Research Centre for infrastructure support.

## Data availability

The data supporting the findings of this study are available in the supplementary material. The GWAS summary statistics of 2083 miRNAs from the Rotterdam Study are publicly available through our miRNomics atlas (www.mirnomics.com) and Zenodo open repository (https://zenodo.org/record/13869398) [88]. Additional data requests can be directed to the corresponding author (M.G).

## Declarations

### Ethics approval and consent to participate

The Rotterdam Study has been approved by the institutional review board (Medical Ethics Committee) of the Erasmus Medical Centre and by the review board of the Netherlands Ministry of Health, Welfare and Sports. Renewal of approval has been conducted every 5 years. Written informed consent was obtained from all participants.
The UK Biobank has approval from the North-West Multi-centre Research Ethics Committee (MREC) as a Research Tissue Bank (RTB) approval. Explicit informed consent was obtained from all participants when they enrolled in the UK Biobank. Access to the UK Biobank was provided through application 52569.

### Consent for publication

Not applicable.

### Competing interests

The authors declare that there are no relationships or activities that might bias, or be perceived to bias, their work.

### Author details

[1]Department of Epidemiology and Biostatistics, School of Public Health, Imperial College London, London, UK. [2]UK Dementia Research Institute, Imperial College London, London, UK. [3]Big Data Institute, Nuffield Department of Population Health, University of Oxford, Oxford, UK. [4]Department of Epidemiology, Erasmus MC University Medical Center Rotterdam, Rotterdam, The Netherlands. [5]Department of Social and Behavorial Sciences, Harvard T.H Chan School of Public Health, Boston, MA, USA. [6]Department of Radiology and Nuclear Medicine, Erasmus MC University Medical Center Rotterdam, Rotterdam, The Netherlands. [7]Institute for Human Development and Potential (IHDP), Agency for Science, Technology and Research (A*STAR), Singapore, Republic of Singapore. [8]Bioinformatics Institute (BII), Agency for Science, Technology and Research (A*STAR), Singapore, Republic of Singapore. [9]Department of Internal Medicine, Erasmus MC University Medical Center Rotterdam, Rotterdam, The Netherlands. [10]Department of Orthopaedics and Sports Medicine, Erasmus MC University Medical Center Rotterdam, Rotterdam, The Netherlands. [11]MRC Centre for Environment and Health, Imperial College London, London, UK. [12]Health Data Research (HDR) UK, Imperial College London, London, UK. [13]BHF Centre for Research Excellence, Imperial College London, London, UK. [14]Framingham Heart Study, Framingham, MA, USA. [15]Population Sciences Branch, National Heart, Lung, and Blood Institute, National Institutes of Health, Bethesda, MD, USA. [16]Department of Mathematics, Imperial College London, London, UK.

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

## 