## [Additional file 3. Peer review history. · Genome Biology]

Review history

First round of review

Reviewer 1

Mustafa et al performed a genome-wide association studies of 2,083 plasma circulating miRNAs measured by a targeted next-generation sequencing in 2,178 participants of the Rotterdam Study to identify miRNA-expression quantitative trait loci. They identified 3,292 associations between 1,289 SNPs and 63 miRNAs, of which 65% were replicated in two independent cohorts. 1,010 unique SNPs and 32 miRNAs had cis associations and 294 unique SNPs and 33 miRNAs had trans associations. Additional independent associations were also discovered based on conditional analysis. The authors performed replication analysis in an independent cohort. They also went on to perform colocalization analysis for the miR eQTLs and mRNA-, protein-, and metabolite-QTLs. Finally, the authors link the miRNA eQTLs with clinical traits via colocalization and Mendelian randomization analysis. This study presents a wealth of data and analysis. Its major drawback is the lack of detailed biological interpretation of their results. Second, does the Rotterdam study have mRNA expression, protein, and metabolite abundance measures in the same subjects where the miRNA measurements were done? If yes, why did not the authors perform colocalization and MR analysis in the multi omics data from the same study participants? If no, the lack of data should be mentioned as a reason to perform colocalization analysis using summary statistics of publicly available datasets.

Other issues are presented below:

-Replication in two independent cohorts is admirable. It is fine to report the SNP-miRNA pair replications; however, the authors should also report the replication on a locus basis, per each miR-eQTL since SNPs are correlated. In other words, let's say in one miR-eQTL, there are 100 SNPs associated with the expression of a single miRNA in cis; yet, in another miR-eQTL, there are 5 SNPs associated with the expression of a single miRNA in cis. If all of the 105 SNPs are present in the replication study and are concordant between the current study and replication study, it looks like 105 SNP-miRNA pairs replicated, whereas the biologically meaningful number is 2.

-The correlation data shown in Fig S3 is impressive. Are the effect size estimates with respect to the same allele in Rotterdam vs other studies? The data is presented in Table S6, but it is not clear if the direction of effect is with respect to the same allele. If yes, this should be indicated; if no, this should be corrected.

-The authors performed colocalization analysis between miR-eQTLs and blood gene expression QTLs to determine the shared genetic signal for miRNAs and either their host or target genes. The authors' miRNA abundance measurements come from plasma, not leukocyte gene expression. As such, the miRNAs they detected could have been secreted by tissues and cells other than those circulating in blood. Why did the authors perform this colocalization analysis only with blood eQTLs? Shouldn't they expand to other tissues that could be the potential source of circulating miRNAs?

-In Table S15, authors' present colocalization of miRNA eQTLs and mRNA QTLs based on the host gene and identify four miRNA eQTLs that are colocalized with the host gene eQTL. If the authors have access to gene expression and miRNA expression from the Rotterdam study, they should plot the correlation of miRNA abundance and its host gene abundance as an additional layer of evidence that they are regulated by the same genetic signal.

-What is the biological interpretation of the colocalization between miRNA eQTLs and protein QTLs? Are the proteins potential targets of miRNAs?

-The section on the colocalization analysis with miRNA eQTLs and metabolite QTLs is underdeveloped. It is not clear if rs174561 is associated with both miR-1908-5p and FADS1 or just miR-1908-5p. There also appear to be associations with lipid metabolites but do they make biological sense? The metabolite IDs presented in column G or Table S20 are meaningless for many of them. What is the biological interpretation of the colocalization between miRNA eQTLs and metabolite QTLs? Are there genes in the metabolite pathways that are potential targets of miRNAs?

Reviewer 2

This is a well-written manuscript describing microRNA eQTL mapping in plasma from a large sample (2178 participants of the Rotterdam study). Replication analyses of identified cis- and trans- eQTL are performed using 2 independent cohorts. Phenome-wide association studies are performed using UK Biobank data to implicate identified genetic regulatory effects on miRNA expression in various clinical diagnoses. I have the following comments:

1. The design for eQTL discovery (at least for discovery of cis-regulatory variants) is a bit atypical. Following the GTEx Consortium (PMID: 29022597; PMID: 32913098), it is now more standard to map cis-eQTL separately by focusing only on SNPs at each gene locus (usually 1MB either side of the gene, but I think 500Kb as used in this study is fine), deriving a P-value and then correcting for the number of genes assayed to provide an FDR (with $FDR < 0.05$ a fairly conventional threshold for cis-eQTL mapping). In this study, the authors have applied the same stringent genome-wide Bonferroni-corrected P-value threshold to their cis-variants as for their trans- variants ($P = 5 \times 10^{-8}$ for genome-wide significance, corrected for 2083 tested miRNAs = $P < 2.4 \times 10^{-11}$). This is too conservative in my view, as I think they will have missed genuine miRNA-cis-eQTL that would have been detected at more conventional P-values / FDR focusing on the cis-regions. An example is the cis-eQTL operating on miR-1908-5p (highlighted in the abstract), with which the authors find interesting clinical associations. Although this eQTL has previously been detected in blood (PMID: 30715214) and other tissues (PMID: 37471622), it does not reach the $P < 2.4 \times 10^{-11}$ threshold and is therefore not initially reported as one of the eQTLs identified in this study (as listed in Supplementary table S3).

2. Related to the above, the eQTL mapping is described in the Methods (line 430) under the title 'genome-wide association studies'. Although this is effectively what the authors have carried out, I think it might be misleading for readers who would more usually consider GWAS as the identification of genetic variants associated with higher level traits or diseases. Even if the authors do not carry out a more targeted initial cis-eQTL screen, as recommended above, I think they should still title this section 'miRNA eQTL mapping' (or similar) instead.

3. Lines 160 -181. Although referenced in the supplementary file, it might be useful for readers if the papers for the datasets used for eQTL, pQTL, and metabolite-QTL overlap are referenced in the main manuscript text (note also that the Vosa et al paper is now published in Nature Genetics).

4. Line 236 - typo - 'Target genes for miR-329-3p were associated with BMI or WHR were identified'

5. Line 265 'The study is currently the most extensive single-site analysis...' I think the authors could be more explicit in stating why this study is the most extensive (it looks like the Nikpay et al paper assayed a large number of microRNA in a smaller sample and the Huan et al paper assayed fewer miRNA in a larger sample).

6. Line 314-319. The authors speculate how variants in the miRNA sequence itself could influence their expression. Toste et al (PMID: 37471622) showed that the eQTL variants in 3 of the microRNA highlighted in this study (miR-1908-5p, miR-4707-3p and miR-323b-3p) are predicted to alter pri-miRNA hairpin secondary structure, with likely effects on their thermostability, so could be cited here.

7. Line 392. The associated cited web tool (www.mirnomics.com) looks like it will be very useful, but when I tried to search for SNPs and miRNA mentioned in the manuscript and listed in supplementary tables, it did not return anything.

Reviewer 3

The study describes a new effort on GWAS of miRNAs in Rotterdam study with replication in additional datasets as well as reciprocal cross replication of previous studies results. This is a major effort to advance our knowledge of the omics data layer variability in human population-based studies, which is important for the scientific community. The manuscript is well-written and is easy to follow, there is also a wealth of well-curated supplementary information and a web-based tool for exploring the results from the study. The quality of statistical analyses attains to the high standards of the journal and the omics research fields. The methods are appropriate, and conclusions are adequate. The presented work represents a significant advancement for the field and is of broad interest to the audience of the Journal as well as a wider audience of biologists.

Major comments

Is there specific correlation structure for the two thousand miRNAs studied? It would be great to have that mention, as it is assumed, they are all independent. This is a very important feature, since the authors delve into pleiotropic loci immediately in the Functional annotations chapter.

The web tool is mentioned apparently only in Conclusion, and it requires wider mention in the manuscript.

The discussion could contain less results and might challenge more the biology of reported discoveries.

In the abstract, it is unclear whether ABO locus and benign colon cancer results are from MR analyses or not.

It would be beneficial to discuss, why only 32 miRNAs from RS provide significant associations out of >2k miRNAs tested. Is the set of such miRNAs similar to the miRNAs reported from studies, where RS was used as replication set in this manuscript. Is there anything specific about these miRNAs as compared to the other without detected associations.

Minor comments

FUMA reference seems to be missing (at least in one place in main text).

Authors' response to reviewers

Reviewer #1

Mustafa et al performed a genome-wide association studies of 2,083 plasma circulating miRNAs measured by a targeted next-generation sequencing in 2,178 participants of the Rotterdam Study to identify miRNA-expression quantitative trait loci. They identified 3,292 associations between 1,289 SNPs and 63 miRNAs, of which 65% were replicated in two independent cohorts. 1,010 unique SNPs and 32 miRNAs had cis associations and 294 unique SNPs and 33 miRNAs had trans associations. Additional independent associations were also discovered based on conditional analysis. The authors performed replication analysis in an independent cohort. They also went on to perform colocalization analysis for the miR eQTLs and mRNA-, protein-, and metabolite-QTLs. Finally, the authors link the miRNA eQTLs with clinical traits via colocalization and Mendelian randomization analysis.

This study presents a wealth of data and analysis. Its major drawback is the lack of detailed biological interpretation of their results.

Response. We thank the reviewer for carefully reading our manuscript and providing constructive feedback. In response, we have revised our manuscript accordingly, including more detailed biological interpretation of our results, as described in the responses below.

Second, does the Rotterdam study have mRNA expression, protein, and metabolite abundance measures in the same subjects where the miRNA measurements were done? If yes, why did not the authors perform colocalization and MR analysis in the multi omics data from the same study participants? If no, the lack of data should be mentioned as a reason to perform colocalization analysis using summary statistics of publicly available datasets.

Response. We thank the reviewer for this important question. We acknowledge that the availability of miRNA and other omics abundance in the same subjects will be very useful in providing another layer of evidence supporting our findings that were mostly based on genetics.

In the Rotterdam Study, there is overlap between plasma miRNA and metabolite data measured by the Metabolon platform (1,087 metabolites, N=512) and the Nightingale platform (179 metabolites, N=975). We have now conducted linear regression analysis using individual-level data of these subjects. In summary, 20,081 miRNA-metabolite pairs can be tested using individual-level data. Nearly 75% of those pairs (15,050) were significant after correcting for multiple testing (FDR<0.05). These pairs were examples of genetic findings coming from summary statistics of publicly available datasets that were in line with individual-level data analysis. We have now reported some of the most significant findings into our results and added the full results into Additional file 1: Table S20.

There is unfortunately no overlap of subjects between miRNA and mRNA expression nor protein abundance in the same subjects of the Rotterdam Study, so the linear regression analysis could not be conducted to support our genetic findings with gene eQTLs and pQTLs. However, we have managed to provide another layer of evidence via colocalisation analysis using genetic association data from independent samples. In this two-sample framework, it is possible to use the largest genetic association data (GWAS or omics-QTLs) on phenotypes or omics of interest to gain higher power in detecting (genetic) associations with miRNA levels.

We have provided revisions in the manuscript as following:

Methods

Population for analysis (Line 577-581)

For a subset of participants, circulatory miRNAs and metabolite abundance measures were measured in the same subjects of the Rotterdam Study. We used this individual-level data to perform linear regression analysis to support our genetic findings. However, at the time of our analysis, there was no overlap between miRNA and gene expression or protein abundance in the Rotterdam Study to assess their correlation.

Linear regression analysis (Line 654-663)

This analysis was performed in the RS-I-4 cohort of the Rotterdam Study to investigate the relationship between miRNA expression levels and metabolite concentrations. Metabolites from the Metabolon platform (1,087 metabolites, 512 participants) and the Nightingale platform (179 metabolites, 975 participants) were used. The analysis involved performing linear regression for each miRNA-metabolite pair while adjusting for: sex, age, smoking status, BMI, red blood cell count, white blood cell count, plate number and well location related to miRNA measurement. The model was fitted using the `lm` function in R, with adjustment to control for the false discovery rate (FDR) using the Benjamini-Hochberg method.

Results

Cross-omics and colocalisation analysis (Line 245-253)

We also conducted linear regression analysis using individual-level data on metabolite levels to support our genetic findings with metabolite-QTLs. In summary, 20,081 miRNA-metabolite pairs with genetic findings can be tested using individual-level data. Of these, nearly 75% (15,050 pairs) were significant after correcting for multiple testing ($FDR < 0.05$). These provided further individual-level data analysis that aligned with our findings from publicly available datasets (Additional file: Table S10). Such analysis could not be performed for gene expression and protein abundance due to lack of data in the Rotterdam Study.

Other issues are presented below:

-Replication in two independent cohorts is admirable. It is fine to report the SNP-miRNA pair replications; however, the authors should also report the replication on a locus basis, per each miR-eQTL since SNPs are correlated. In other words, let's say in one miR-eQTL, there are 100 SNPs associated with the expression of a single miRNA in cis; yet, in another miR-eQTL, there are 5 SNPs associated with the expression of a single miRNA in cis. If all of the 105 SNPs are present in the replication study and are concordant between the current study and replication study, it looks like 105 SNP-miRNA pairs replicated, whereas the biologically meaningful number is 2.

Response. We thank the reviewer for this comment. We consider discovered miR-eQTLs in the Rotterdam Study and replicated miR-eQTLs as the input for genomic-loci mapping. This mapping was conducted using FUMA (1), with several parameters described in our Methods section, including the strategy to avoid double counting of signals due to correlation between SNPs. Our results per locus basis is presented in Additional file: Table S10. We have now moved the parameter details of FUMA (particularly to address correlation between SNPs) from Additional file to our main Methods section.

Following this comment, we did a further check and found that 19 out of 22 loci harbored miR-eQTLs have been replicated across cohorts. We have now made this clear in our Results and added relevant information on how many miRNAs have replicated associations within each locus into Additional file: Table S10. In this table, we added a new column F: *n_miRNAs_with_replicated_associations* that indicate the biologically meaningful number of miRNAs with replicated miR-eQTLs across cohorts, whereas the new column G listed the name of miRNAs.

We also highlight the two most pleiotropic loci associated with several miRNAs, one in chr14:100655022-101244293 predominantly consisting of *cis* miR-eQTLs, and one in chr9:136128546-136296530 predominantly consisting of *trans* miR-eQTLs, in our main text.

We have now revised our manuscript accordingly in the Methods and Results sections to make this clear for our readers.

Methods:

Functional annotation of miR-eQTLs (Line 623-630)

We used SNP2GENE process in FUMA (1) to annotate miR-eQTLs into genomic risk loci and mapped them to genes according to their position. Independent significant miR-eQTLs were defined as those with $P < 2.4 \times 10^{-11}$ in the discovery or those replicated in independent cohorts and in moderate LD with each other ($r^2 < 0.6$). LD calculation was referenced based on 1000 Genomes phase 3 panel. These SNPs were further clumped to lead SNPs ($r^2 < 0.1$). Genomic risk loci were then defined based on the lead SNPs when they overlap with a maximum distance of 250kb between LD blocks.

Results:

Line 143-150

We performed mapping for discovered and replicated miR-eQTLs across studies. These miR-eQTLs were mapped using FUMA (Methods) (1) to identify the genomic loci regulating miRNA expression in plasma. These miR-eQTLs were mapped into 22 genomic loci, of which 11 loci were pleiotropic, i.e., linked to the level of multiple miRNAs (Additional file 1: Table S10). Among 22, 19 loci harbored miR-eQTLs that have been replicated across cohorts. One noteworthy highly pleiotropic locus was identified on chr14:100655022-101244293, known as 14q32 miRNA cluster, regulating 23 miRNAs, predominantly as *cis*-miR-eQTLs (Additional file 2: Fig. S4).

Line 159-160

Another highly pleiotropic locus was on chr9:136128546-136296530 mapped to ABO and other genes (Additional file 1: Table S10) and regulated 18 miRNAs.

Additional File 1

Table S10. List of 22 loci, miRNAs, and corresponding genes. Nineteen loci harbored miRNAs with replicated miR-eQTLs across cohorts. Within each locus, the miRNAs with replicated association across independent cohorts were shown (column F-G)

-The correlation data shown in Fig S3 is impressive. Are the effect size estimates with respect to the same allele in Rotterdam vs other studies? The data is presented in Table S6, but it is not clear if the direction of effect is with respect to the same allele. If yes, this should be indicated; if no, this should be corrected.

Response. Yes, the alleles were harmonized so that the effect estimates correspond to the same alleles in both discovery and replication cohorts. We have now explained this in our Methods section and provided this information on the caption for Figure S3, as following:

Methods

Line 603-606

We harmonized the alleles so that the effect estimates between discovery and replication cohorts corresponded to the same effect alleles. Replication was defined when the associations between SNP and miRNA were Bonferroni-significant in an independent cohort with a concordant direction of effect.

Additional file 2

Supplementary Figure S3

... For replication, we harmonized the alleles so that the effect estimates between discovery and replication cohorts corresponded to the same effect alleles.

-The authors performed colocalization analysis between miR-eQTLs and blood gene expression QTLs to determine the shared genetic signal for miRNAs and either their host or target genes. The authors' miRNA abundance measurements come from plasma, not leukocyte gene expression. As such, the miRNAs they detected could have been secreted by tissues and cells other than those circulating in blood. Why did the authors perform this colocalization analysis only with blood eQTLs? Shouldn't they expand to other tissues that could be the potential source of circulating miRNAs?

Response. We thank the reviewer for this important question. Our analysis used the largest sample size for blood gene expression in eQTLGen (2) to improve the power of detecting shared genetic signals. We agree that miRNAs measured in plasma could come from different tissues and cells; thus, using tissue-level expression data will benefit our analysis. We have therefore conducted colocalisation analysis using tissue-level gene expression data in GTex v8 (3) to find shared genetic signals with host and nearby genes (as proxy for miRNAs) across different tissues. As the reviewer pointed out, this analysis might suggest potential tissue source(s) of circulating miRNAs. The results of this analysis are now combined into the Additional file: Table S15. Moreover, we have revised our manuscript in order to highlight some examples of our findings and their biological relevance.

Methods:

Line 644-647

To explore this, we sought overlaps between replicated miR-eQTLs and gene expression (eQTLs) in whole blood (eQTLGen) (2) and across 49 tissues (GTEx v8) (4), protein (pQTLs)(5-9), and metabolite-QTLs (mQTLs)(5-7) (Additional file 2: Methods S5).

Additional file 1

Table S15. Colocalisation analysis between plasma miRNAs and gene expression in whole blood (eQTLGen) and across tissues (GTEx). Since we only had miR-eQTLs in plasma, host or nearby gene eQTLs were used as proxies for miRNAs across tissues. Results are presented for those with evidence of colocalisation (PP.H4.abf > 0.7). Most likely candidate causal SNPs (with highest SNP.PP.H4) are shown.

Results

Line 199-214

We also conducted a colocalisation analysis between miRNAs and gene expression across 49 tissues using the GTEx dataset (4). The colocalisation was conducted when cis-miR-eQTLs were found to be at least significantly associated with gene expression ($P < 0.05$) in each tissue. We screened for 64 miRNAs reported in our replication analysis, then tested for 20,979 associations (46 miRNAs and 909 genes across 49 tissues). We identified 450 associations with $PP.H4 > 0.7$ between 30 miRNAs and 106 genes (Additional file: Table S15). For example, we additionally found evidence of a shared genetic signal between miR-584-5p and its host gene (SH3TC2) in the lung and between miR-335-5p and its host gene (MEST) in 13 other tissues, including brain, artery, and adipose tissues. While colocalisation analysis with tissue-level gene expression data allowed us to identify shared genetic signals with host or nearby genes and, to some extent, indicate that the miRNAs expressed in plasma might also be expressed or act in those tissues, it is of importance to do such analysis using tissue-specific miRNA expression when such data is available in a large cohort. This tissue-wide analysis may help to pinpoint the potential source of circulating miRNAs and elucidate their tissue specificity.

-In Table S15, authors' present colocalization of miRNA eQTLs and mRNA QTLs based on the host gene and identify four miRNA eQTLs that are colocalized with the host gene eQTL. If the authors have access to gene expression and miRNA expression from the Rotterdam study, they should plot the correlation of miRNA abundance and its host gene abundance as an additional layer of evidence that they are regulated by the same genetic signal.

Response. We agree that testing for correlation between miRNA and host gene abundance could provide additional layer of evidence for regulation by the same genetic signals.

Unfortunately, there is no overlap of participants with miRNA and gene expression data in the Rotterdam Study, so it is not possible to do this analysis. We have added this as a limitation of our study in Discussion in the revised manuscript.

Discussion

Line 491-496

Our colocalisation analyses showed shared genetic signals between miRNAs and their host or target genes, proteins, and metabolites. We provide another layer of evidence for correlation between miRNA and metabolite abundance using individual-level data in the Rotterdam Study. Similarly, this analysis could also be done with their respective host or target genes abundance. However, gene expression data is not available for the same participants in the Rotterdam Study at the time of the analysis.

-What is the biological interpretation of the colocalization between miRNA eQTLs and protein QTLs? Are the proteins potential targets of miRNAs?

Response. Colocalisation indicates shared (potentially) causal signals. Colocalisation with host genes indicates that miRNAs might be co-expressed with their host or nearby genes due to co-regulation by the same regulatory region. Colocalization between cis-miR-eQTLs with pQTLs of distant genes might indicate that those genes are being targeted by miRNAs.

In our analysis, we found that at least several proteins with shared genetic regulations are potential targets for miRNAs such as miR-127-3p, miR-136-5p, miR-431-5p, and miR-433-5p with *DLK1* and *SEMG2*, and miR-625-5p with Alpha-(1,6)-fucosyltransferase (*FUT8*) (as shown in Additional file 1: Table S17). We have now highlighted these findings in our Results. We further provided an example of miR-625-5p and *FUT8*, which showed colocalisation with gene eQTLs and pQTLs. This example supports our hypothesis that colocalisation with mRNA and protein can help to determine if a putative target based on bioinformatic predictions is a bona fide miRNA target. In cases where there is no evidence of MTI with miRNA, we hypothesize that a third factor (such as an upstream regulatory mechanism) could explain the observed colocalisation. These results have now been added to the Results section. In the Discussion section, we also elaborate on our hypothesis on the possible explanation of shared signals between miRNAs and protein QTLs.

If miRNA-target interactions exist, colocalization could provide evidence at both gene expression and protein levels. Since miRNAs are known to inhibit the translation of mRNAs into proteins, analyzing both mRNA and protein levels is common practice in miRNA research to confirm whether a predicted target based on bioinformatics is a true miRNA target. For instance, miR-625-5p and *FUT8* demonstrated colocalization with eQTLs and pQTLs. While those miRNA-target pairs with evidence of colocalization were among predicted miRNA-target interaction (MTI) in TargetScan (10), none has been experimentally validated in previous studies according to miRTarBase (11) at the time of this analysis. Our study highlights the importance of both in silico approaches for identifying shared genetic signals and potential miRNA-target interactions among numerous candidate genes, as well as the need for experimental validation to confirm these interactions.

Results

Line 226-238

In our analysis, we found that at least several proteins with shared genetic regulations that their related genes were potential targets for miRNAs, such as miR-127-3p, miR-136-5p, miR-431-5p, and miR-433-5p with *DLK1* and *SEMG2*, as well as miR-625-5p with Alpha-(1,6)-fucosyltransferase (Additional file 1: Table S17). To note, those miRNA-target pairs were among

predicted miRNA-target interaction (MTI) in TargetScan (8), with none have been validated experimentally in previous studies according to miRTarBase (9), at the time of this analysis. If there is miRNA-target interaction, colocalisation could provide evidence at both gene expression and protein levels, as miRNAs are expected to repress translation of mRNAs into protein. As an example, miR-625-5p and *FUT8* demonstrated colocalisation with gene eQTLs and pQTLs. Our analysis highlighted the equal importance of in silico and experimental studies to elucidate shared genetic signals underlying potential MTI and validation of those interactions.

Discussion

Line 381-392

Our integrative QTL analysis showed that miR-eQTLs colocalize with gene expression and protein QTLs of their target genes, supporting the role of miRNAs in gene regulation and translational repression. Since some target genes tend to be clustered to miRNAs according to their function (12,13), these shared miR-eQTLs might have biological relevance. Cis-miR-eQTLs that overlap with trans-mRNA-eQTLs might point to the downstream regulatory effect from miRNAs to their (direct or indirect) target genes. On the other hand, for cis-mRNA-eQTLs overlap with trans-miR-eQTLs, the effect might be going from the genes to miRNAs, pointing to bidirectional interaction between miRNAs and target genes as a feedback mechanism (14,15). However, when trans-miR-eQTL overlap with trans-mRNA-eQTLs without evidence of miRNA and target gene interaction, a third factor, such as upstream regulatory mechanism, may have contributed to simultaneous changes in miRNA and gene expression.

-The section on the colocalization analysis with miRNA eQTLs and metabolite QTLs is underdeveloped. It is not clear if rs174561 is associated with both miR-1908-5p and *FADS1* or just miR-1908-5p. There also appear to be associations with lipid metabolites but do they make biological sense? The metabolite IDs presented in column G or Table S20 are meaningless for many of them. What is the biological interpretation of the colocalization between miRNA eQTLs and metabolite QTLs? Are there genes in the metabolite pathways that are potential targets of miRNAs?

Response. We appreciate the reviewer for this feedback. We found that rs174561 is associated with both miR-1908-5p ($P=7.66 \times 10^{-8}$) in our analysis and *FADS1* according to eQTLGen data ($P=3.27 \times 10^{-310}$). The difference in magnitude of *P-values* is due to much larger sample size in eQTLGen ($N \approx 30,000$) compared to our study ($N \approx 2,178$). Despite this, we found evidence of colocalisation between expression of miR-1908-5p and *FADS1* in the circulation (PP.H4.abf = 0.8), with most likely candidate causal variant being rs102275 (Additional file: Table S15). The presence of shared genetic signal between miRNA and host gene raised question whether the effect identified between miR-1908-5p and benign neoplasm of colon has been driven by *FADS1* rather than miR-1908-5p. Therefore, we conducted multivariable MR to address this question. Briefly, we treated both miR-1908-5p and *FADS1* as exposures in the MR analysis and found that the effect was significant for miR-1908-5p but not *FADS1*, despite the larger sample size in eQTLGen compared to our study. This finding indicates that the effect of miR-1908-5p on the disease risk is independent of its host gene, or in other words, the miRNA is the key player here. This finding is also supported by colocalisation analysis where we found evidence of shared signal between benign neoplasm of colon and miR-1908-5p (PP.H4.abf=0.58), but not *FADS1* (PP.H4.abf=0.008) (Fig. 7). In our view, such multivariable analysis is very important, and

we strongly recommend it to be performed in any future analysis to exclude the possibility of the host gene being the key player rather than the miRNA of interest, given that both often share genetic regulation.

To elucidate pathways mediating the effect of miR-1908-5p on benign neoplasm of colon, we conducted further MR analysis looking at the effect of miR-1908-5p on the metabolite levels using Metabolon platform (16), which annotated and measured metabolite levels across different classes. This analysis showed many metabolites that might act as mediators between miR-1908-5p and benign neoplasm of colon belonging to lipid classes. Our linear regression analysis between miR-1908-5p presented in earlier part of manuscript (Results: cross-omics and colocalisation) also identified lipid metabolites associated with this miRNA measured using both the Metabolon and Nightingale platforms.

The association with lipid found here aligns with our previous study reporting association between miR-1908-5p with LDL-cholesterol, total cholesterol, triglyceride, and HDL-cholesterol (17). Our current and previous study both highlight the relevance of lipid pathways for miR-1908-5p. Furthermore, here we show how this pathway correlates miR-1908-5p and benign neoplasm of colon.

We have revised the Results and Discussion sections to make our findings clearer for the readers and highlight the importance of this analysis. We explain this in the context of findings related to miR-1908-5p and more generally for future studies on miRNAs and human health.

Results

Line 318-347

We found evidence of colocalisation between expression of miR-1908-5p and *FADS1* gene in the circulation (PP.H4.abf = 0.8), with most likely candidate causal variant being rs102275 (Additional file: Table S15). The presence of shared genetic signal between miRNA and host gene raised question whether the effect identified between miR-1908-5p and benign neoplasm of colon has been driven by *FADS1* rather than miR-1908-5p. We therefore conducted multivariable MR to address this question. Colocalisation analysis suggested shared causal variant between benign neoplasm of colon and miR-1908-5p (PP H4 = 0.58), but not for *FADS1* (PP H4 = 0.008) (Fig. 7). Our multivariable MR, where we treated both miR-1908-5p and *FADS1* as exposures, showed an attenuation of effect of miR-1908-5p (effect estimate=-0.010, $P=7.78 \times 10^{-06}$), but no effect of *FADS1* ($P=0.67$). Overall, these suggested that miR-1908-5p was the putative causal factor that altered disease risk, with the effect likely being independent of the host gene.

To elucidate pathways mediating the effect of miR-1908-5p on benign neoplasm of colon, we conducted further MR analysis looking at the effect of miR-1908-5p on metabolite levels using the Metabolon platform (16), which annotated and measured metabolite levels across different classes. This analysis identified 102 metabolites affected by miR-1908-5p (Additional file 1: Table S31, Additional file 2: Fig. S9), of which 12 metabolites also found to affect the risk of disease – all belonging to lipid class (Additional file 1: Table S32, Additional file 2: Fig. S10). Multivariable MR showed 1-palmitoyl-2-linoleoyl-GPE (16:0/18:2) remaining as the only significant metabolite ($P=4.67 \times 10^{-03}$) (Additional file 2: Fig. S10), in line with the results of MR-BMA (Additional file 1: Table S33). Finally, mediation analysis showed that 1-palmitoyl-2-linoleoyl-GPE (16:0/18:2) mediated 49% (3.14%-95.31%) of the total effect of miR-1908-5p (Additional file 1: Table S34). Further analysis showed no significant effect of miR-1908-5p

when adjusting for the 12 candidate metabolites, indicating that the overall effect of miRNA may drive through all those metabolites (Additional file 1: Table S35). The association with lipid found here aligns with our previous study reporting association between miR-1908-5p with LDL-cholesterol, total cholesterol, triglyceride, and HDL-cholesterol (17).

Discussion

Line 472-474

Our current and previous study both highlight the relevance of lipid pathways for miR-1908-5p. Furthermore, here we show how this pathway links miR-1908-5p and benign neoplasm of colon.

Line 480-490

In this study, we showed the importance of disentangling the effect of miRNA from host genes. There are several approaches that can be used to fulfil this. If both miRNA and host gene expression data is available in the same participants, conditional analysis could be performed with adjustment on the expression of host genes for any associations identified for miRNAs, as demonstrated previously (18). If the data is not available for the same participants, genetic association data could be used within multivariable framework, such as through multivariable MR analysis. Our example for miR-1908-5p, *FADS1*, and benign neoplasm of colon serves as example of the latter. This approach should be carried out in future research to exclude the possibility of the host gene being the key player rather than the miRNA of interest, given that both often have shared genetic regulation.

Reviewer #2

This is a well-written manuscript describing microRNA eQTL mapping in plasma from a large sample (2178 participants of the Rotterdam study). Replication analyses of identified cis- and trans- eQTL are performed using 2 independent cohorts. Phenome-wide association studies are performed using UK Biobank data to implicate identified genetic regulatory effects on miRNA expression in various clinical diagnoses. I have the following comments:

1. The design for eQTL discovery (at least for discovery of cis-regulatory variants) is a bit atypical. Following the GTEx Consortium (PMID: 29022597; PMID: 32913098), it is now more standard to map cis-eQTL separately by focusing only on SNPs at each gene locus (usually 1MB either side of the gene, but I think 500Kb as used in this study is fine), deriving a P-value and then correcting for the number of genes assayed to provide an FDR (with $FDR < 0.05$ a fairly conventional threshold for cis-eQTL mapping). In this study, the authors have applied the same stringent genome-wide Bonferroni-corrected P-value threshold to their cis-variants as for their trans- variants ($P = 5 \times 10^{-8}$ for genome-wide significance, corrected for 2083 tested miRNAs = $P < 2.4 \times 10^{-11}$). This is too conservative in my view, as I think they will have missed genuine miRNA-cis-eQTL that would have been detected at more conventional P-values / FDR focusing on the cis-regions. An example is the cis-eQTL operating on miR-1908-5p (highlighted in the abstract), with which the authors find interesting clinical associations. Although this eQTL has previously been detected in blood (PMID: 30715214) and other tissues (19), it does not reach the $P < 2.4 \times 10^{-11}$ threshold and is therefore not initially reported as one of the eQTLs identified in this study (as listed in Supplementary table S3).

Response. We thank the reviewer for their constructive comments. We agree that using the FDR-based threshold for cis-miR-eQTLs identification could potentially highlight more cis-miR-eQTLs. With a less conservative threshold for miR-eQTL discovery, we found more trans-miR-eQTLs, which, in turn, had lower replication rates, indicating these trans-miR-eQTLs might not be genuine signals (Additional file 1: Table S7). Thus, we decided to use a more conservative threshold in our discovery analysis while also trying to validate the findings from two previous studies, by Nikpay et al. and Huan et al., at their defined study significance levels (Additional file 2: Supplementary Figure 2). In the end, we reported a set of replicated miR-eQTLs for 64 miRNAs, including associations for 20 miRNAs that did not reach our original discovery threshold of $P < 2.4 \times 10^{-11}$. This set included cis variant for miR-1908-5p that have been detected in previous studies. We considered this set as the most robust findings across multiple cohorts. We have now revised the manuscript to make this more clear.

We also relaxed the cis-miR-eQTLs threshold for genetic instrument selection in our PheWAS and MR analysis using an $FDR < 0.1$ to cover more miRNAs in the downstream analysis. This allowed us to include miR-1908-5p, among others, in our downstream analysis. This FDR-threshold could provide an alternative set of cis-miR-eQTLs to be used by our readers. We have now added these cis-miR-eQTLs in the supplementary data (Additional file: Table S23), and we will release all the data through our publicly available web tool (www.mirnomics.com). We have provided description on the impact of relaxing significance threshold for our miR-eQTLs identification in the revised manuscript.

Results:

Line 132-139

We finally reported all miR-eQTLs replicated across cohorts, including those that were not initially discovered using a stringent threshold ($P < 2.4 \times 10^{-11}$) in our discovery. These were considered most robust findings including 4,310 replicated associations for 64 miRNAs (Additional file 1: Table S8). These included associations for 20 miRNAs that originally did not reach our study significance threshold. An example of these were cis-variants for miR-1908-5p, such as rs174561, which was previously reported as miR-eQTL in plasma and other tissues (19,20).

Line 274-278

We implemented FDR-based threshold for every miRNA (in the region spanning 500kb either side of the miRNA position) in our PheWAS and MR analyses to enable identifying more instruments and covering more miRNAs. The summary statistics of FDR-significant cis-miR-eQTLs are provided in Additional file 1: Table S23, and the full results are available through our web tool (www.mirnomics.com).

Discussion

Line 499-504

At the genome-wide significance threshold, we observed lower replication rates for miR-eQTLs, with an overall decrease of 22.6%. This decline was much more pronounced in the replication of trans (32.2%) compared to cis-miR-eQTLs (9.2%). This indicates that many trans-signals detected at the conventional genome-wide threshold contained non-genuine signals that were less replicable, consistent with a previous study (18).

Line 448-456

We used an FDR-based threshold for every miRNA (+/-500kb of the miRNA position) in the PheWAS and MR analysis to enable identifying more instruments and covering more miRNAs. This decision was made based on several reasons: 1) cis miR-eQTLs are considered biologically relevant, and FDR-based methods are commonly used in cis-eQTL discovery (2), 2) the main concern with a relaxed P-value threshold in the MR analysis is the possibility for weak instruments bias, whereas such bias tends to be towards the null (false negative) in the setting of two-sample MR, as implemented in our study, 3) we replicated our PheWAS and MR analysis using independent datasets to avoid false positive findings.

2. Related to the above, the eQTL mapping is described in the Methods (line 430) under the title 'genome-wide association studies'. Although this is effectively what the authors have carried out, I think it might be misleading for readers who would more usually consider GWAS as the identification of genetic variants associated with higher level traits or diseases. Even if the authors do not carry out a more targeted initial cis-eQTL screen, as recommended above, I think they should still title this section 'miRNA eQTL mapping' (or similar) instead.

Response. We fully agree with this statement and we have revised the title in the Methods section and provide further clarification within the text, as follows:

Methods

Line 582

Identification and mapping of miRNA expression quantitative trait loci

Line 587-590

Identification of genetic variants associated with miRNA expression in plasma, or so-called miRNA expression quantitative trait loci (miR-eQTLs), both acting in proximity (cis) or distant (trans), was performed through genome-wide association studies (GWAS) for each of 2,083 miRNAs.

3. Lines 160 -181. Although referenced in the supplementary file, it might be useful for readers if the papers for the datasets used for eQTL, pQTL, and metabolite-QTL overlap are referenced in the main manuscript text (note also that the Vosa et al paper is now published in Nature Genetics).

Response. We thank the reviewer for this comment. We have now added the citation for those papers in the main text and updated with the latest reference as follows:

Methods

Line 644-647

To explore this, we sought overlaps between replicated miR-eQTLs and gene expression (eQTLs) in whole blood (eQTLGen) (2) and across 49 tissues (GTex v8) (4), protein (pQTLs)(5-9), and metabolite-QTLs (mQTLs)(5-7) (Additional file 2: Methods S5).

4. Line 236 - typo - 'Target genes for miR-329-3p were associated with BMI or WHR were identified'

Response. We have now corrected this sentence as follows:

Results:

Line 307-310

.. Through an in-silico search of target genes using TargetScan v7.2(10) and miRTarBase(11), eighty-two predicted and eighteen validated target genes associated with BMI or WHR were found for miR-543. Likewise, 43 predicted and 58 validated target genes associated with BMI or WHR were identified for miR-329-3p.....

5. Line 265 'The study is currently the most extensive single-site analysis...' I think the authors could be more explicit in stating why this study is the most extensive (it looks like the Nikpay et al paper assayed a large number of microRNA in a smaller sample and the Huan et al paper assayed fewer miRNA in a larger sample).

Response. We have now made this more clear in the Discussion.

Line 350-357

The study is currently the most extensive single-site analysis of over 2,000 circulating miRNAs in 2,178 individuals from the Rotterdam Study cohort, followed by replication in two independent cohorts. Our study has expanded the coverage of miRNAs compared to the

previous study covering 280 circulatory miRNAs in whole blood in the Framingham Heart Study (18). Moreover, the sample size in our study is three times larger than the previous study by Nikpay et al. (19), which measured 2,083 circulatory miRNA levels using the same miRNA profiling method.

6. Line 314-319. The authors speculate how variants in the miRNA sequence itself could influence their expression. Toste et al (PMID: 37471622) showed that the eQTL variants in 3 of the microRNA highlighted in this study (miR-1908-5p, miR-4707-3p and miR-323b-3p) are predicted to alter pri-miRNA hairpin secondary structure, with likely effects on their thermostability, so could be cited be here.

Response. We thank the reviewer for this comment and have now added this publication into our revised manuscript.

Discussion

Line 407-409

Examples were shown in previous study, such as by Toste et al. reporting miR-eQTLs of miR-1908-5p, miR-4707-3p and miR-323b-3p were predicted to alter corresponding pri-miRNA hairpin secondary structure (20).

7. Line 392. The associated cited web tool (www.mirnomics.com) looks like it will be very useful, but when I tried to search for SNPs and miRNA mentioned in the manuscript and listed in supplementary tables, it did not return anything.

Response. This is indeed our plan to release our full results on the web tool, making them publicly available upon acceptance of this manuscript.

Reviewer #3

The study describes a new effort on GWAS of miRNAs in Rotterdam study with replication in additional datasets as well as reciprocal cross replication of previous studies results. This is a major effort to advance our knowledge of the omics data layer variability in human population-based studies, which is important for the scientific community. The manuscript is well-written and is easy to follow, there is also a wealth of well-curated supplementary information and a web-based tool for exploring the results from the study. The quality of statistical analyses attains to the high standards of the journal and the omics research fields. The methods are appropriate, and conclusions are adequate. The presented work represents a significant advancement for the field and is of broad interest to the audience of the Journal as well as a wider audience of biologists.

Response. We thank the reviewer for providing constructive feedback. We have revised our manuscript accordingly as described in more details below.

Major comments

Is there specific correlation structure for the two thousand miRNAs studied? It would be great to have that mention, as it is assumed, they are all independent. This is a very important feature, since the authors delve into pleiotropic loci immediately in the Functional annotations chapter.

Response. We acknowledge the reviewer's comment. The correlation between miRNAs is an important feature when interpreting the pleiotropy of identified loci. Our pairwise correlation analysis across all 64 identified miRNAs with genetic findings resulted in a median absolute correlation coefficient of 0.14 (interquartile range (IQR): 0.23).

In addition, the median absolute correlation coefficient between miRNAs that shared genetic regulation appears to be higher than the correlation across all studied miRNAs. For example, the pleiotropic locus in chr9:136128546-136296530 associated with 18 miRNAs, showed a higher median absolute correlation coefficient of 0.61 (IQR: 0.16). Another example in chr14:100655022-101244293, regulating 23 miRNAs, showed a higher median absolute correlation coefficient of 0.35 (IQR: 0.23). Despite this, there remain three miRNAs, namely miR-345-5p, miR-411-3p, miR-433-3p, in this cluster which do not correlate with any other miRNAs (absolute correlation coefficient < 0.3 with any miRNAs in the same locus. These observations suggest that one locus can be truly pleiotropic by regulating multiple independent miRNAs. These results are now added into the Results section and presented in a new Additional File: Fig.S6.

Both genetic and environmental factors influence human phenotypes and the correlation between them (21). Phenotypic correlation can arise due to several reasons, such as shared genetic and environmental determinants or the presence of causal relationships between phenotypes. While genome-wide genetic correlation may often resemble phenotypic correlation, the genetic contribution on a locus basis could be different (22). We acknowledge the importance of careful interpretation when looking at pleiotropic effect for miRNAs, considering that these miRNAs might often be correlated with each other. Future studies could aim to disentangle the pleiotropic effect from phenotypic correlation between miRNAs or leverage these correlations to improve the power in genetic discovery.

We have now added correlation plots of miRNAs with genetic signals in two most pleiotropic as examples in Additional File: Fig.S6 and further details and explanations regarding the correlation between miRNAs and careful interpretation of pleiotropy of miR-eQTLs loci.

Results

Line 143-165

We performed mapping for discovered and replicated miR-eQTLs across studies. These miR-eQTLs were mapped using FUMA (Methods) (1) to identify the genomic loci regulating miRNA expression in plasma. These miR-eQTLs were mapped into 22 genomic loci, of which 11 loci were pleiotropic, i.e., linked to the level of multiple miRNAs (Additional file 1: Table S10). One noteworthy highly pleiotropic locus was identified on chr14:100655022-101244293, known as 14q32 miRNA cluster, regulating 23 miRNAs, predominantly as cis-miR-eQTLs (Additional file 2: Fig. S4). While pairwise phenotypic correlation analysis across all 64 miRNAs with genetic findings resulted in median absolute correlation coefficient of 0.14 (interquartile range (IQR): 0.23), the absolute correlation coefficient between miRNAs in this locus appeared to be higher (median: 0.35, IQR: 0.23) (Additional file 1: Fig. S5). Nevertheless, there remains three miRNAs in this cluster which do not correlate with any other miRNAs (absolute correlation coefficient < 0.3) in the same locus, namely miR-345-5p, miR-411-3p, miR-433-3p (Additional file 1: Fig. S4). These observations may indicate that one locus could be truly pleiotropic by regulating multiple independent miRNAs.

Another highly pleiotropic locus was on chr9:136128546-136296530 mapped to ABO and other genes (Additional file 1: Table S10) and regulated 18 miRNAs. This locus contained shared trans-miR-eQTLs for several well-known miRNAs, such as miR-10, let-7, and miR-30 families (Additional file 1: Table S10), contributing to 84 out of 143 conditionally independent trans-associations (Additional file 1: Table S4). The median absolute correlation coefficient between these 18 miRNAs was 0.61 (IQR:0.16) (Additional file 1: Fig. S4).

Discussion

Line 422-432

The pleiotropic loci identified in our study are associated with multiple miRNAs, some of which are phenotypically (their expression levels) correlated to each other. Both genetics and environmental factor influence human phenotypes and the correlation between them(21). Phenotypic correlation could arise due to several reasons, such as shared genetic and environmental determinants or the presence of causal relationships between phenotypes. While genome-wide genetic correlation could be similar to phenotypic correlation in many instances, the genetic contribution on a locus-basis could be different (22). We acknowledge that one needs to be careful when looking at pleiotropic effect for miRNAs. Future studies could try to disentangle the true pleiotropic effect from the phenotypic correlation between miRNAs or rather to take advantage of these correlation to improve power in genetic discovery.

Additional file 2

Supplementary Figure S6. A. Number of miRNA pairs with genetic findings (64x43 miRNAs pairs tested), categorised by their absolute correlation coefficient. b. Correlation plots between miRNAs within two most pleiotropic loci.

a.

b.

chr14:100655022-101244293 (23 miRNAs)

chr9:136128546-136296530 (18 miRNAs)

Conclusio

manuscript.

Response. Following the reviewer’s suggestion, we have now mentioned the web tool both in the Abstract and Results section of our revised manuscript, as follows:

Abstract

Line 59-60

... All the results are publicly accessible through the miRNomics atlas (www.mirnomics.com).

Results

Line 107-109

The results described here and the full summary statistics are accessible through the miRNomics atlas (www.mirnomics.com).

Line 274-279

The summary statistics of the FDR-significant cis-miR-eQTLs are provided in Additional file 1: Table S23, and the full results are available through our web tool (www.mirnomics.com).

The discussion could contain less results and might challenge more the biology of reported discoveries.

Response. We have now revised the discussion to contain less results and instead include more biological interpretation of our findings in the Discussion as follows:

Low replicability of trans-miR-eQTLs (Line 499-504)

At the genome-wide significance threshold, we observed lower replication rates for miR-eQTLs (overall decrease by 22.6%), with much pronounced decline in replication of trans (32.2%) than cis (9.2%) (Additional file 1: S7), indicating many trans-signals detected at conventional genome-wide threshold contained non-genuine signals that were less replicable, in line with a previous study (18).

Caution on the use trans-miR-eQTLs for MR analysis (Line 506-511)

Trans miR-eQTLs were found to be more pleiotropic by being associated with other omics QTLs. This made them unsuitable for instrumenting miRNAs due to the risk of horizontal pleiotropy which could violate the MR assumptions. It is therefore recommended to use cis miR-eQTLs, although caution remains needed, given that they are also often shared with host or nearby genes, as shown in our example on miR-1908-5p and *FADS1*.

Relevance of lipid pathways for miR-1908-5p (Line 472-474)

Our current and previous study both highlight the relevance of lipid pathways for miR-1908-5p. Furthermore, here we show how this pathway links miR-1908-5p and benign neoplasm of colon.

Relevance of genetic variants in miRNA sequence (Line 407-411)

Examples were shown in previous study, such as by Toste et al. reporting miR-eQTLs of miR-1908-5p, miR-4707-3p and miR-323b-3p that were predicted to alter corresponding pri-miRNA hairpin secondary structure (20). Our previous work has shown that variants residing in miRNA-related sequence have functional relevance (23).

To disentangle the effect of miRNA from host gene (Line 480-490)

In this study, we showed the importance of disentangling the effect of miRNA from host genes. There are several approaches that can be used to fulfill this. If both miRNA and host gene expression data is available in the same participants, conditional analysis could be performed with adjustment on the expression of host genes for any associations identified for miRNAs, as demonstrated previously (18). If the data is not available for the same participants, genetic association data could be used within multivariable framework, such as through multivariable MR analysis. Our example for miR-1908-5p, *FADS1*, and benign neoplasm of colon serves as example of the latter. This approach should be carried out in future research to exclude the

possibility of the host gene being the key player rather than the miRNA of interest, given that both often have shared genetic regulation.

Correlation between miRNAs and pleiotropy of loci (Line 422-432)

The pleiotropic loci identified in our study are associated with multiple miRNAs, some of which are phenotypically (their expression levels) correlated to each other. Both genetics and environmental factor influence human phenotypes and the correlation between them(21). Phenotypic correlation could arise due to several reasons, such as shared genetic and environmental determinants or the presence of causal relationships between phenotypes. While genome-wide genetic correlation could be similar to phenotypic correlation in many instances, the genetic contribution on a locus basis could be different (22). We acknowledge that one needs to be careful when looking at pleiotropic effect for miRNAs. Future studies could try to disentangle the true pleiotropic effect from the phenotypic correlation between miRNAs or rather to take advantage of these correlation to improve power in genetic discovery.

In the abstract, it is unclear whether ABO locus and benign colon cancer results are from MR analyses or not.

Response. The finding for benign neoplasm of colon is from our MR analysis, whereas for ABO locus was not. We have now removed the part related to ABO and replaced this with mentioning the web tool in Abstract, as follows:

Abstract

Line 57-60

In the MR analysis, we found protective causal effect of miR-1908-5p on the risk of benign colon neoplasm and showed that this effect is independent of its host gene (FADS1). All the results are publicly accessible through our miRNomics atlas (www.mirnomics.com).

It would be beneficial to discuss, why only 32 miRNAs from RS provide significant associations out of >2k miRNAs tested. Is the set of such miRNAs similar to the miRNAs reported from studies, where RS was used as replication set in this manuscript. Is there anything specific about these miRNAs as compared to the other without detected associations.

Response. We acknowledge the reviewer's suggestion. In total, we identified 3,292 associations for 63 miRNAs within the Rotterdam Study cohort only. We listed difference in characteristics of these 63 miRNAs compared to those without detected associations:

- 1) This subset of miRNAs is more heritable than the rest of studied miRNAs as indicated by higher heritability estimates (mean:0.12 for 63 miRNAs vs mean:0.08 for 2,083 miRNAs).
- 2) This subset is expressed at higher levels in the biological samples used in this study (plasma), meaning that we can measure these well-expressed miRNAs reasonably better than the rest.

Within our cohort, we also performed quality control which included assessment whether miRNAs are well-expressed in plasma. Of 2,083 miRNAs, 591 (nearly 28%) were found to be well-expressed in plasma, based on the ratio between their mean expression and standard deviation in the whole samples. Of the 63 identified miRNAs with genetic associations in our study, 47 miRNAs (75%) were among the well-expressed miRNAs in plasma. We have now

added these explanations in our Methods and Results sections and provided information on well-expressed miRNAs in Additional file 1: Table S1.

There was limited overlap of miRNAs with genetic findings from other studies, i.e., 20/143 miRNAs, and 15/57 miRNAs overlapping with those reported in Nikpay et al. (19) and Huan et al. (18), respectively. Nevertheless, a set of 9 miRNAs were common across the three studies, despite the difference in biological sample type (plasma vs whole blood) and miRNA profiling method (sequencing vs qPCR) to those in Huan et al. (18). We discuss several considerations when attempting to replicate miR-eQTLs across independent studies in our Discussion.

While the associations identified in plasma for those circulatory miRNAs remains valid, our observations underline the need for tissue-specific profiling of miRNAs that are not well-expressed in the circulation, particularly when studying specific diseases of interest. We also note that the heritability estimates are relatively low for many miRNAs. Although sample size remains a limitation, particularly for small genetic effects, this may imply that miRNAs are under stronger selective constraints that limit their variability (24), such that common variants do not show strong effects. This explanation is also presented in the Discussion.

We have revised our manuscript accordingly.

Methods:

Line 561-565

Quantification of miRNA expression was based on counts per million (CPM). Log₂ transformation of CPM was used as standardisation and adjustment for the total reads within each sample. MiRNAs with log₂ CPM 50% values above the lower limit of quantification (LLOQ). Out of 2,083 miRNAs, 591 were well-expressed in the samples (Additional file 1: Table S1).

Additional file 1

Table S1. The list of 2,083 miRNAs characterised in this study. Among 2,083 miRNAs, 591 were well-expressed in our cohort.

Results:

Line 181-186

We found that 63 miRNAs with significant findings in our discovery analysis was on average more heritable than the rest of studied miRNAs as indicated by higher heritability estimates (mean:0.12 for 63 miRNAs vs mean:0.08 for 2,083 miRNAs). Of these 63 miRNAs, 47 miRNAs were among the well-expressed miRNAs in plasma (Methods), meaning that we could measure these miRNAs reasonably better than the rest.

Discussion:

Line 368-377

Our results showed 63 out of the 2,083 studied miRNAs (approximately 3% of all miRNAs and 10% of the well-expressed miRNAs) have common variants associated with their plasma levels, which could be a relatively small proportion compared to those identified for messenger RNA eQTLs. Although sample size remains a limitation, this may imply that miRNAs have stronger selective constraints that limit their variability (24), such that common variants do not show strong effects. Our heritability analysis revealed the modest effect of genetic variants on

plasma miRNA levels, as also shown by the small variation explained by miR-eQTLs, which could act as a mechanism to maintain biological function during evolution.

Line 497-499

We should underline several aspects to be considered when attempting to replicate miR-eQTLs across studies. First, we found that fewer trans- were replicated than cis-miR-eQTLs, as observed in the large eQTL analysis as well (2).

Line 512-521

Second, the concordant direction with those reported by Nikpay et al. (19) suggested that the type of biological sample and miRNA profiling method could have an effect. The lower replication rate in the Framingham Heart Study is likely due to differences in the type of sample (whole blood vs plasma), as previously reported (25), and the miRNA profiling method (qPCR vs targeted RNA-seq). Third, one should consider any systematic difference in participants' characteristics across studies. This study came from a population-based cohort which makes the findings more generalisable. Other studies were in obese individuals (19) or enriched for a specific disease (26), making it particularly useful for investigating the relevant disease but not for a wide range of complex traits and disorders.

Minor comments

FUMA reference seems to be missing (at least in one place in main text).

Response. We thank the reviewer for highlighting this. We have now added this citation in the main text.

Methods:

Line 620-621

The web-based tool Functional Mapping and Annotation (FUMA) was used to annotate miR-eQTLs (1).

Line 623-624

We used SNP2GENE process in FUMA (1) to annotate miR-eQTLs into genomic risk loci and mapped them to genes according to their position.

Results:

Line 143-145

We performed mapping for replicated miR-eQTLs across studies. These miR-eQTLs were mapped using FUMA (Methods) (1) to identify the genomic loci regulating miRNA expression in plasma.

References

- (1) Watanabe K, Taskesen E, Van Bochoven A, Posthuma D. Functional mapping and annotation of genetic associations with FUMA. *Nature communications*. 2017; 8 (1): 1-11.
- (2) Vösa U, Claringbould A, Westra H, Bonder MJ, Deelen P, Zeng B, et al. Large-scale cis-and trans-eQTL analyses identify thousands of genetic loci and polygenic scores that regulate blood gene expression. *Nature genetics*. 2021; 1-11.
- (3) GTEx Consortium. Genetic effects on gene expression across human tissues. *Nature*. 2017; 550 (7675): 204-213.
- (4) GTEx Consortium. The Genotype-Tissue Expression (GTEx) pilot analysis: Multitissue gene regulation in humans. *Science*. 2015; 348 (6235): 648-660.
- (5) Shin S, Fauman EB, Petersen A, Krumsiek J, Santos R, Huang J, et al. An atlas of genetic influences on human blood metabolites. *Nature genetics*. 2014; 46 (6): 543-550.
- (6) Kettunen J, Demirkan A, Würtz P, Draisma HH, Haller T, Rawal R, et al. Genome-wide study for circulating metabolites identifies 62 loci and reveals novel systemic effects of LPA. *Nature communications*. 2016; 7 (1): 1-9.
- (7) Elsworth B, Lyon M, Alexander T, Liu Y, Matthews P, Hallett J, et al. The MRC IEU OpenGWAS data infrastructure. *BioRxiv*. 2020; .
- (8) Sun BB, Maranville JC, Peters JE, Stacey D, Staley JR, Blackshaw J, et al. Genomic atlas of the human plasma proteome. *Nature*. 2018; 558 (7708): 73-79.
- (9) Folkersen L, Gustafsson S, Wang Q, Hansen DH, Hedman ÅK, Schork A, et al. Genomic and drug target evaluation of 90 cardiovascular proteins in 30,931 individuals. *Nature metabolism*. 2020; 2 (10): 1135-1148.
- (10) Agarwal V, Bell GW, Nam J, Bartel DP. Predicting effective microRNA target sites in mammalian mRNAs. *elife*. 2015; 4 e05005.
- (11) Huang H, Lin Y, Li J, Huang K, Shrestha S, Hong H, et al. miRTarBase 2020: updates to the experimentally validated microRNA–target interaction database. *Nucleic acids research*. 2020; 48 (D1): D148-D154.
- (12) Gamazon ER, Innocenti F, Wei R, Wang L, Zhang M, Mirkov S, et al. A genome-wide integrative study of microRNAs in human liver. *BMC genomics*. 2013; 14 (1): 395.
- (13) Huan T, Rong J, Liu C, Zhang X, Tanriverdi K, Joehanes R, et al. Genome-wide identification of microRNA expression quantitative trait loci. *Nature communications*. 2015; 6 (1): 1-9.
- (14) Mustafa R, Ghanbari M, Evangelou M, Dehghan A. An enrichment analysis for cardiometabolic traits suggests non-random assignment of genes to microRNAs. *International journal of molecular sciences*. 2018; 19 (11): 3666.
- (15) Sakaue S, Hirata J, Maeda Y, Kawakami E, Nii T, Kishikawa T, et al. Integration of genetics and miRNA–target gene network identified disease biology implicated in tissue specificity. *Nucleic acids research*. 2018; 46 (22): 11898-11909.

- (16) Melling GE, Flannery SE, Abidin SA, Clemmens H, Prajapati P, Hinsley EE, et al. A miRNA-145/TGF- β 1 negative feedback loop regulates the cancer-associated fibroblast phenotype. *Carcinogenesis*. 2018; 39 (6): 798-807.
- (17) Aguda BD, Kim Y, Piper-Hunter MG, Friedman A, Marsh CB. MicroRNA regulation of a cancer network: consequences of the feedback loops involving miR-17-92, E2F, and Myc. *Proceedings of the National Academy of Sciences*. 2008; 105 (50): 19678-19683.
- (18) Chen Y, Lu T, Pettersson-Kymmer U, Stewart ID, Butler-Laporte G, Nakanishi T, et al. Genomic atlas of the plasma metabolome prioritizes metabolites implicated in human diseases. *Nature genetics*. 2023; 1-10.
- (19) Ghanbari M, Sedaghat S, De Looper HW, Hofman A, Erkeland SJ, Franco OH, et al. The association of common polymorphisms in mi R-196a2 with waist to hip ratio and mi R-1908 with serum lipid and glucose. *Obesity*. 2015; 23 (2): 495-503.
- (20) Nikpay M, Beehler K, Valsesia A, Hager J, Harper M, Dent R, et al. Genome-wide identification of circulating-miRNA expression quantitative trait loci reveals the role of several miRNAs in the regulation of cardiometabolic phenotypes. *Cardiovascular research*. 2019; 115 (11): 1629-1645.
- (21) Toste CC, O'Donovan MC, Bray NJ. Mapping microRNA expression quantitative trait loci in the prenatal human brain implicates miR-1908-5p expression in bipolar disorder and other brain-related traits. *Human molecular genetics*. 2023; 32 (20): 2941-2949.
- (22) Pickrell JK, Berisa T, Liu JZ, Ségurel L, Tung JY, Hinds DA. Detection and interpretation of shared genetic influences on 42 human traits. *Nature genetics*. 2016; 48 (7): 709-717.
- (23) Shi H, Mancuso N, Spendlove S, Pasaniuc B. Local genetic correlation gives insights into the shared genetic architecture of complex traits. *The American Journal of Human Genetics*. 2017; 101 (5): 737-751.
- (24) Mustafa R, Ghanbari M, Karhunen V, Evangelou M, Dehghan A. Phenome-wide association study on miRNA-related sequence variants: the UK Biobank. *Human Genomics*. 2023; 17 (1): 104.
- (25) Rotival M, Siddle KJ, Silvert M, Pothlichet J, Quach H, Quintana-Murci L. Population variation in miRNAs and isomiRs and their impact on human immunity to infection. *Genome biology*. 2020; 21 1-31.
- (26) Westra H, Peters MJ, Esko T, Yaghootkar H, Schurmann C, Kettunen J, et al. Systematic identification of trans eQTLs as putative drivers of known disease associations. *Nature genetics*. 2013; 45 (10): 1238-1243.
- (27) Yao C, Joehanes R, Johnson AD, Huan T, Liu C, Freedman JE, et al. Dynamic role of trans regulation of gene expression in relation to complex traits. *The American Journal of Human Genetics*. 2017; 100 (4): 571-580.
- (28) Shah R, Tanriverdi K, Levy D, Larson M, Gerstein M, Mick E, et al. Discordant expression of circulating microRNA from cellular and extracellular sources. *PloS one*. 2016; 11 (4): e0153691.
- (29) Akiyama S, Higaki S, Ochiya T, Ozaki K, Niida S, Shigemizu D. JAMIR-eQTL: Japanese genome-wide identification of microRNA expression quantitative trait loci across dementia types. *Database*. 2021; 2021 (2021): baab072.

Second round of review

Reviewer 1

The authors thoroughly addressed the issues I raised in the initial review.

Reviewer 2

The authors have satisfactorily responded to my comments.

Reviewer 3

The authors have made a great effort to address the reviewers' comments. The manuscript clarity and details are improved. I am fully satisfied with the proposed new version